# Large language model validity via enhanced conformal prediction methods

**John J. Cherian**
Department of Statistics
Stanford University
jcherian@stanford.edu

**Isaac Gibbs**
Department of Statistics
Stanford University
igibbs@stanford.edu

**Emmanuel J. Candès**
Department of Statistics
Department of Mathematics
Stanford University
candes@stanford.edu

## Abstract

We develop new conformal inference methods for obtaining validity guarantees on the output of large language models (LLMs). Prior work in conformal language modeling identifies a subset of the text that satisfies a high-probability guarantee of correctness. These methods work by filtering claims from the LLM's original response if a scoring function evaluated on the claim fails to exceed a threshold calibrated via split conformal prediction. Existing methods in this area suffer from two deficiencies. First, the guarantee stated is not conditionally valid. The trustworthiness of the filtering step may vary based on the topic of the response. Second, because the scoring function is imperfect, the filtering step can remove many valuable and accurate claims. We address both of these challenges via two new conformal methods. First, we generalize the conditional conformal procedure of Gibbs et al. (2023) in order to adaptively issue weaker guarantees when they are required to preserve the utility of the output. Second, we show how to systematically improve the quality of the scoring function via a novel algorithm for differentiating through the conditional conformal procedure. We demonstrate the efficacy of our approach on biography and medical question-answering datasets.

## 1 Introduction

Large language models (LLMs) are a breakthrough in machine learning. In addition to their extraordinary performance on natural language processing benchmarks, LLMs such as ChatGPT and Gemini are now used by hundreds of millions of users around the world [24]. But even though these models match or even surpass human performance on an increasingly complex and diverse set of tasks, their reliability remains in doubt. For example, LLMs often confidently hallucinate facts that do not exist, and can generate toxic outputs that may offend or discriminate [22]. This "misalignment" between user goals and model behavior hinders LLM deployment in settings where the potential for AI assistance appears highest, e.g., legal work or customer service interaction [31].

Since an LLM output is not always trustworthy, a growing body of work aims to quantify uncertainty regarding a given output's validity. While there are many approaches to this problem [28, 4, 9, 18], this paper considers a particularly popular method for black-box uncertainty quantification: conformal inference [30, 2, 3]. Conformal inference provides a generic methodology for transforming the predictions of any modeling procedure into valid prediction sets that are guaranteed to contain the true outcome with high probability. Several recent papers have applied conformal inference to define a set of LLM responses that contains at least one factual response with high probability [3, 17, 25, 32]. But while generating a candidate set of outputs may be a reasonable strategy in some question-answering problems, it is not a generalizable approach for the diverse and unstructured tasks faced in real-world deployment.

38th Conference on Neural Information Processing Systems (NeurIPS 2024).

| Output | Conformal Factuality | Our Method |
|---|---|---|
| | Fixed level: 90% | Adaptive level: 63% |

The shingles vaccine is typically recommended for adults aged 50 and older. The vaccine is given in two doses, with the second dose administered 2 to 6 months after the first dose. It is currently recommended that individuals receive the shingles vaccine once in their lifetime. However, it is always best to consult with a healthcare provider for personalized recommendations.

The shingles vaccine is typically recommended for adults ~~aged 50 and older. The vaccine is given in two doses, with the second dose administered 2 to 6 months after the first dose. It is currently recommended that individuals receive the shingles vaccine once in their lifetime. However, it is always best to consult with a healthcare provider for personalized recommendations.~~

The shingles vaccine is typically recommended for adults aged 50 and older. The vaccine is given in two doses, with the second dose administered 2 to 6 months after the first dose. ~~It is currently recommended that individuals receive the shingles vaccine once in their lifetime.~~ However, it is always best to consult with a healthcare provider for personalized recommendations.

**Figure 1.** The left panel displays the output of GPT-3.5-Turbo for the prompt "How often is a shingles vaccine required?" The first filtered output (center) is calibrated using the frequency score (see Appendix E.1) and the marginally valid conformal factuality method of Mohri and Hashimoto [21] at a fixed level of 90%. The second filtered output (right) is calibrated using a score obtained via our conditional boosting procedure (Section 3.3) at a level of 63%, which is chosen and calibrated using our adaptive method (Section 3.2) to approximately ensure that at least 70% of the claims are retained. Both filtered outputs are guaranteed to include no false claims with the stated probability.

More recently, Mohri and Hashimoto [21] propose to forgo sets of LLM outputs and instead utilize conformal inference to filter out invalid components of the LLM response. At a high level, given an LLM generation parsed into a set of distinct sub-claims, their method censors all sub-claims for which a pre-defined scoring function lies below some threshold. Mohri and Hashimoto [21] then show how to calibrate this threshold such that the retained claims are factual with high probability.

While these methods represent a promising step towards usable guarantees for LLM outputs, they are not yet practical. One limitation is that the guarantee attained by previous methods only holds *marginally* over a random test prompt. The true probability of output correctness may then vary substantially based on the prompt's characteristics. For example, we show in Section 4 that the probability of output correctness (even after applying the conformal factuality method) is substantially lower for responses whose subjects are likely to be underrepresented in the model's training corpus. Second, existing methods remove too many claims to be practically useful. Recall that we remove sub-claims for which some pre-defined score falls below a calibrated threshold. If this score is perfect, only false claims will be censored. In practice, however, these scores are only weakly correlated with the ground truth. As Figure 1 demonstrates, a high probability factuality guarantee can require the removal of a significant proportion of the generated text.[1] The conformal guarantee is not useful if the filtered response has limited value for the end-user.

## 1.1 Summary of contributions

In this subsection, we will preview and summarize our results. A more complete description of our theory and experimental setup is deferred to Sections 3 and 4.

As in prior literature on conformal language modeling, we will assume the existence of an annotated calibration set of $n$ i.i.d. prompt-response-claim-annotation tuples, $\{(P_i, R_i, \mathbf{C}_i, \mathbf{W}_i)\}_{i=1}^n$. The vector $\mathbf{C}_i$ is obtained by using an LLM to parse the response into a list of scorable sub-claims, while $\mathbf{W}_i$ might correspond to human verification of the underlying factuality of each claim. To simplify notation, we will refer to these tuples using the shorthand, $\mathbf{D}_i$.

At first glance, the twin goals we have outlined for this paper, improved conditional validity *and* enhanced quality of filtered outputs, appear to be irreconcilable. Indeed, prior work establishes that precise conditional guarantees in black-box uncertainty quantification require larger prediction set sizes, i.e., smaller filtered outputs [5, 29]. We contribute two methods to mitigate this trade-off, thus enabling the practical application of conformal prediction to LLMs.

Our first method, which we call **conditional boosting**, allows for the automated discovery of superior claim scoring functions via differentiation through the conditional conformal algorithm of Gibbs et al. [10]. Automated conformal score improvement was introduced by Stutz et al. [27]; their paper shows how to minimize conformal prediction set size in a classification setting by differentiating through the marginally valid split conformal algorithm. As we show, however, in Section 4, optimiz-

---

[1]We note that when conformal prediction is applied to more typical supervised learning tasks, this problem is equivalent to the challenge of prediction set "inefficiency," i.e. large prediction set size.

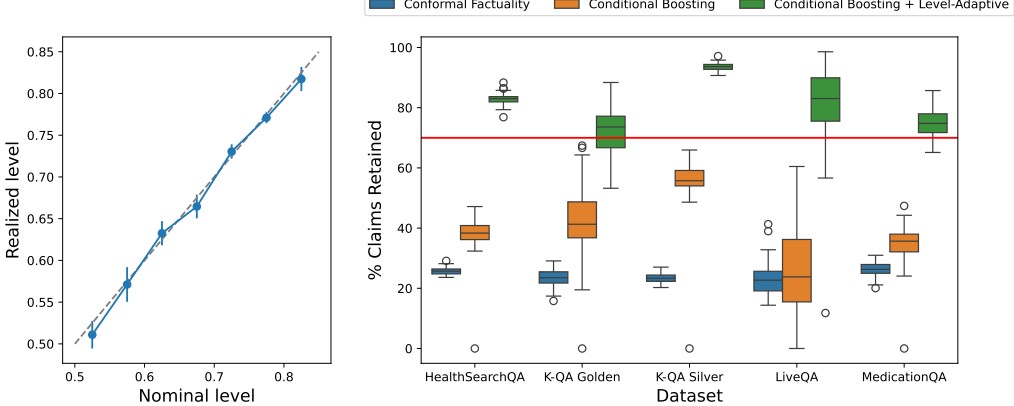

**Figure 2.** Empirical demonstration of our methods. The panels display results for our conditional boosting and level-adaptive methods. We aim to issue outputs with 0 factual errors, and for the latter method, we choose the level with the objective of retaining at least 70% of the original claims in the prompt. The left panel compares the binned nominal probabilities of factuality reported by our method against the realized probability of factuality for data points belonging to each bin. These probabilities are estimated using 500 test points over 100 calibration-test splits. The plotted bins, which are also given as inputs to our method, are $[0.5, 0.55], [0.55, 0.6], \dots, [0.8, 0.85]$. Finally, the right-hand panel displays the claim retention obtained with unboosted scores (blue), boosted scores (orange), and boosted scores + level-adaptive CP (green). The first two methods are implemented at a fixed error rate of $\alpha = 0.1$. Boxplots in this panel show the distribution of retained claims for 100 calibration-test splits with each containing 2354 calibration points and 500 test points.

ing the score function subject only to a marginal coverage constraint can lead to poor conditional properties.

Optimizing through the conditional conformal algorithm is not straightforward. Our key technical contributions are a proof that (under mild assumptions) the cutoff output by the conditional conformal method is differentiable and a computationally efficient method for computing this derivative. By running gradient descent using this algorithm we discover new scores that enable greater claim retention.

The right panel of Figure 2 demonstrates the efficacy of our method. Here, we use boosting to learn an optimal linear combination of four candidate scoring functions. We compare the learned, boosted scores (orange) against a baseline method (blue) that uses the "frequency" scoring method developed by Mohri and Hashimoto [21]. As the figure shows, the boosted score allows for higher claim retention across all datasets (mean retention of 39% vs. 24% for the boosted vs. unboosted scores).

Our second method, which we call **level-adaptive conformal prediction**, allows the validity of the conformal output to depend on characteristics of the queried prompt. In our LLM experiments, we adapt the level, i.e., the claimed probability of correctness, individually to each prompt in order to ensure that issued outputs retain at least 70% of the original set of sub-claims. For example, in Figure 1, we prompt GPT-3.5-Turbo to output a response to a question from the **MedicationQA** dataset [6]. Outputting a filtered response that achieves the stated factuality criterion with probability 90% requires near-complete censorship, but by relaxing the level to 63% using our method, we can preserve almost the entire response.

Given that we are now issuing an output-adaptive probability of correctness, it is crucial that our issued probability is *calibrated*. Calibration requires that the true probability of correctness matches the issued one. For example, if a weather forecaster claims that there is a 70% chance of rain, their forecast is calibrated if it actually rains for 70% of the days on which a 70% forecast is issued.

Figure 2 displays the advantages of our approach to this problem. First, the left panel of Figure 2 verifies that the level-adaptive probabilities we report are empirically well-calibrated. Second, the right panel of Figure 2 quantitatively demonstrates the improved claim retention of our method and verifies that for each dataset included in the **MedLFQA** benchmark [13] our level-adaptive conformal prediction retains at least 70% of the original output's claims in most examples. Finally, by

combining our level-adaptive and conditional boosting methods, we retain most claims *and* output non-trivial guarantees of response factuality; the left panel shows that the issued probabilities vary between 50 and 85%. By contrast, while the fixed level method guarantees a 90% probability of correctness, the method retains very little of the original LLM output.

To emphasize that these results are accompanied by formal guarantees, we preview one instantiation of our theory here. Since it is well-known that exact conditional guarantees in conformal inference are impossible to achieve without strong distributional assumptions [5, 29], we present an interpretable alternative: group-conditional calibration.[2] For example, in this dataset, we might group questions by medical area or data provenance; we would then hope to show that across health conditions or data sources, the claimed probability of factuality matches the true probability of factuality.

Equation (1), which follows from Theorem 3.2, presents one guarantee that our method can satisfy. Here, we denote the (random) output of our data-adaptive level function by $\alpha_{n+1}$ and our filtered set of claims by $\hat{F}(\mathbf{C}_{n+1})$. Our method then satisfies the following guarantee simultaneously over groups $G \in \mathcal{G}$ (e.g., prompt topic, data provenance) and some discretization of $[0, 1]$ given by the sub-intervals $I$ (e.g., all sub-intervals with endpoints belonging to $\{0, 0.1, \ldots, 1\}$),

$$\mathbb{P}\left(\hat{F}(\mathbf{C}_{n+1}) \text{ is factual} \mid \alpha_{n+1} \in I, P_{n+1} \in G\right) = \mathbb{E}[\alpha_{n+1} \mid \alpha_{n+1} \in I, P_{n+1} \in G]. \qquad (1)$$

More concretely, (1) establishes that the issued probabilities are well-calibrated in the following sense: among similar prompts, the outputs that we claim to be factually correct with probability, say, between 70 and 80% will be *actually* factual between 70 and 80% of the time. In Section 3.1, we show how our framework can be adapted to guarantee that the LLM's response satisfies other alignment targets beyond factual accuracy.

The remainder of the paper is outlined as follows. In Section 2, we introduce the formal notation of our paper and contextualize our approach by reviewing related work in conformal inference. Section 3 then presents our new methodology for conformal language modeling. We first generalize the conditional conformal procedure of Gibbs et al. [10] to obtain high-probability control of arbitrary monotone risks. We then state and give intuition for the key technical results underpinning our level-adaptive and boosting methods. Section 4 outlines synthetic experiments displaying the improvements of our approach over existing methods, gives a more detailed description of the experiment presented in Figure 2 above, and presents another experiment that filters short biographies output by an LLM.

We release a filtered version of the **MedLFQA** benchmark that removes some non-health-related prompts, the generated and parsed text used to run our experiments, as well as the notebooks used to produce the figures in this paper at github.com/jjcherian/conformal-safety. We also update our Python package for conditional conformal inference to support level-adaptive conformal prediction. This package is available to download at github.com/jjcherian/conditional-conformal and can be installed from PyPI.

## 2   Background and related work

### 2.1   Conformal prediction with conditional guarantees

Split conformal prediction provides a generic procedure for transforming the outputs of any blackbox model into valid prediction sets [23, 30]. Let $\{(X_i, Y_i)\}_{i=1}^{n+1}$ be a set of covariate-ground truth pairs where $X_{n+1}$ denotes a test value for which we would like to output a response. Then, split conformal outputs a prediction set $\hat{C}(X_{n+1})$ such that

$$\mathbb{P}(Y_{n+1} \in \hat{C}(X_{n+1})) = 1 - \alpha, \qquad (2)$$

for any user-specified value $\alpha \in (0, 1)$.

While powerful, the guarantee given in (2) only holds on-average over the test value. Critically, in the LLM context we consider, this means that methods calibrated using split conformal prediction run the risk of displaying systematically worse performance on the most safety-critical examples.

---

[2]Equation (1) closely resembles a measure of calibration error known in the theoretical computer science literature as multicalibration [12]. Formally, however, multicalibration is defined conditionally on the calibration dataset, while the expectation in (1) is over both the calibration and test points.

This problem is addressed in Gibbs et al. [10], which introduces a novel target for obtaining coverage conditional on covariate information. In their work, they observe that exact covariate-conditional coverage can also be expressed as a marginal guarantee over any measurable function $f \in \mathcal{F}$, i.e.,

$$\mathbb{P}(Y_{n+1} \in \hat{C}(X_{n+1}) \mid X_{n+1}) = 1 - \alpha$$
$$\Leftrightarrow$$
$$\mathbb{E}[f(X_{n+1}) \cdot (\mathbb{1}\{Y_{n+1} \in \hat{C}(X_{n+1})\} - (1-\alpha))] = 0 \qquad \text{for all measurable } f.$$

Since exact distribution-free covariate-conditional coverage requires the analyst to issue vacuous prediction sets [29], Gibbs et al. [10] design a prediction set that satisfies the same marginal guarantee over a user-specified function class $\mathcal{F}$, i.e.,

$$\mathbb{E}[f(X_{n+1}) \cdot (\mathbb{1}\{Y_{n+1} \in \hat{C}(X_{n+1})\} - (1-\alpha))] = 0 \qquad \text{for all } f \in \mathcal{F}. \qquad (3)$$

To make this guarantee concrete, consider the case where $\mathcal{F} := \{(\mathbb{1}\{X \in G\})^\top \beta \mid \beta \in \mathbb{R}^{|\mathcal{G}|}\}$ for some set of subgroups $\mathcal{G}$. Then, (3) is exactly equivalent to group-conditional coverage, i.e., $\mathbb{P}(Y_{n+1} \in \hat{C}(X_{n+1}) \mid X_{n+1} \in G) = 1 - \alpha$ for all $G \in \mathcal{G}$.

To understand their construction, let $S(X, Y) \in \mathbb{R}$ denote a conformity score that measures how well $Y$ "conforms" to an estimate of its value. A typical choice in the regression setting is $S(X, Y) = |Y - \hat{\mu}(X)|$ for some fixed regression function $\hat{\mu}(\cdot)$; in the classification setting, we might choose $S(X, Y) = 1/\hat{\pi}(Y \mid X)$ for some estimated conditional probabilities $\hat{\pi}$.

Gibbs et al. [10] estimate a high-probability upper bound for these scores by fitting an augmented quantile regression in which the unknown test score, $S(X_{n+1}, Y_{n+1})$ is replaced by an imputed value $S$. Formally, let $\ell_\alpha(r) := (1-\alpha)[r]_+ + \alpha[r]_-$ denote the pinball loss. Then, they define

$$g_S = \operatorname{argmin}_{g \in \mathcal{F}} \frac{1}{n+1} \sum_{i=1}^{n} \ell_\alpha(S(X_i, Y_i) - g(X_i)) + \frac{1}{n+1} \ell_\alpha(S - g(X_{n+1})), \qquad (4)$$

and output the prediction set given by $\hat{C}(X_{n+1}) := \{y : S(X_{n+1}, y) \leq g_{S(X_{n+1}, y)}(X_{n+1})\}$.

Since $\hat{C}(X_{n+1})$ can be mildly conservative, Gibbs et al. [10] also define a smaller randomized prediction set, $\hat{C}_{\text{rand.}}(X_{n+1})$ that achieves exact coverage. We will typically prefer to work with this set, but defer a detailed definition of its construction to the Appendix. As the following theorem shows, this randomized set achieves the guarantee stated in (3).

**Theorem 2.1** (Proposition 4 of Gibbs et al. [10]). *Let $\mathcal{F} = \{\Phi(X)^\top \beta : \beta \in \mathbb{R}^d\}$ denote any finite dimensional linear class. Assume that $\{(X_i, S_i)\}_{i=1}^{n+1}$ are exchangeable and that solutions to (4) and its dual are computed symmetrically on the input data. Then, for all $f \in \mathcal{F}$,*

$$\mathbb{E}[f(X_{n+1})(\mathbb{1}\{S_{n+1} \in \hat{C}_{rand.}(X_{n+1})\} - (1-\alpha))] = 0.$$

### 2.2 Conformal factuality

As discussed in the introduction, prediction sets are not suitable for many LLM use-cases. As an alternative, Quach et al. [25] and Mohri and Hashimoto [21] use conformal inference to develop filtering methods that remove false claims from an LLM's output. We will focus specifically here on the work of Mohri and Hashimoto [21] since this is most similar to the new methods that we will propose in this article. Recall that $\mathbf{C}_i = \{C_{ij}\}_{j=1}^{k_i}$ denotes the claims made in an LLM's response and $\mathbf{W}_i = \{W_{ij}\}_{j=1}^{k_i}$ denotes binary variables for which $W_{ij} = 1$ indicates that the claim is true.

Mohri and Hashimoto [21] aim to output a set of filtered claims, $\hat{F}(\mathbf{C}_i) \subseteq \mathbf{C}_i$, that contains no errors with high probability, i.e.,

$$\mathbb{P}\left(\exists C_{(n+1)j} \in \hat{F}(\mathbf{C}_{n+1}) \text{ such that } W_{(n+1)j} = 0\right) \leq \alpha. \qquad (5)$$

To achieve this target, Mohri and Hashimoto [21] define a scoring function $p(P_i, C_{ij}) \in \mathbb{R}$ that takes a prompt and claim as input and summarizes the LLM's internal confidence in the claim. Here,

larger values of $p(P_i, C_{ij})$ indicate that the LLM believes that $C_{ij}$ is more likely to be true. Then, Mohri and Hashimoto [21] set

$$\hat{F}(\mathbf{C}_i) := F(\mathbf{C}_i; \hat{\tau}) = \{C_{ij} : p(P_i, C_{ij}) \geq \hat{\tau}\},$$

where $\hat{\tau}$ is the $\frac{\lceil (1-\alpha)(n+1) \rceil}{n+1}$-quantile of the conformity scores,

$$S(\mathbf{C}_i, \mathbf{W}_i) = \inf\{\tau \mid \forall C_{ij} \in \hat{F}(\mathbf{C}_i; \tau), W_{ij} = 1\}. \tag{6}$$

Mirroring the proof of split conformal prediction [23], Theorem 1 of [21] shows that if $\{(P_i, R_i, \mathbf{C}_i, \mathbf{W}_i)\}_{i=1}^{n+1}$ are exchangeable, this method satisfies (5).

## 3 Methods

### 3.1 Generalization to alternative targets

Our first contribution in this paper will be to generalize the conditional framework of Gibbs et al. [10] to accommodate generic LLM alignment tasks. To do so, we extend the conformal risk control framework [3, 15] to provide high-probability control of a monotone loss with *conditional* guarantees.

More concretely, suppose that we are given a loss function $L(\hat{F}(\mathbf{C}_i), \mathbf{W}_i)$ that measures the quality of the filtered output $\hat{F}(\mathbf{C}_i)$ relative to the ground truth $\mathbf{W}_i$. Our goal will be to ensure that $\mathbb{P}(L(\hat{F}(\mathbf{C}_{n+1}), \mathbf{W}_{n+1}) \leq \lambda) \geq 1 - \alpha$, for some user-specified tolerance $\lambda > 0$. For example, in the prior section $L(\cdot, \cdot)$ was the binary indicator that $\hat{F}(\mathbf{C}_{n+1})$ contains a false claim. More generally, $L(\cdot, \cdot)$ could measure the presence of toxic or discriminatory content in the response.

To incorporate general losses into the conditional conformal framework of Gibbs et al. [10], we require two assumptions. First, we assume that the method is always permitted to abstain from issuing a response and thus $L(\emptyset, \cdot) = 0$. Second, we assume that the loss is monotone. Namely, for any sets of claims $\hat{F}_1(\mathbf{C}_i) \subseteq \hat{F}_2(\mathbf{C}_i)$, it must be the case that $L(\hat{F}_1(\mathbf{C}_i), W_i) \leq L(\hat{F}_2(\mathbf{C}_i), W_i)$.

With these assumptions, we extend the calibration procedure of Gibbs et al. [10] as follows. Let $X_i = X(P_i, R_i)$ denote a set of features computed using the prompt and response. Matching (6), we define $p(P_i, C_{ij})$ to be a score measuring the quality of claim $C_{ij}$. We define the filtered set of claims by $\hat{F}(\mathbf{C}_i) = F(\mathbf{C}_i; \hat{\tau}) := \{C_{ij} : p(P_i, C_{ij}) \geq \hat{\tau}\}$, and the conformity score via

$$S(\mathbf{C}_i, \mathbf{W}_i) := \inf\{\tau \mid L(F(\mathbf{C}_i; \tau), \mathbf{W}_i) \leq \lambda\}.$$

Our two assumptions (monotonicity of the loss and $L(\emptyset, \cdot) = 0$) imply that $S(\mathbf{C}_i, \mathbf{W}_i)$ is well-defined and equal to the minimum loss-controlling threshold. Finally, we set

$$g_S = \mathrm{argmin}_{g \in \mathcal{F}} \frac{1}{n+1} \sum_{i=1}^{n} \ell_\alpha(S(\mathbf{C}_i, \mathbf{W}_i) - g(X_i)) + \frac{1}{n+1} \ell_\alpha(S - g(X_{n+1})), \tag{7}$$

and filter claims at the cutoff $\hat{\tau}(X_{n+1}) = \sup\{S \mid S \leq g_S(X_{n+1})\}$. Similar to the prediction sets of Gibbs et al. [10], $\hat{\tau}(X_{n+1})$ can be conservative, and, thus, we prefer to use a randomized analog $\hat{\tau}_{\mathrm{rand.}}(X_{n+1})$; its formal definition can be found in Appendix A. As the following theorem shows, this randomized cutoff satisfies the desired guarantee.

**Theorem 3.1.** *Let $\mathcal{F} = \{\Phi(X)^\top \beta : \beta \in \mathbb{R}^d\}$ denote any finite dimensional linear class. Assume that $\{\mathbf{D}_i\}_{i=1}^{n+1}$ are exchangeable, that solutions to (7) and its dual are computed symmetrically in the input data, $L(\cdot, \cdot)$ is monotone in its first argument, and $L(\emptyset, \cdot) = 0$. Then, for all $f \in \mathcal{F}$,*

$$\mathbb{E}[f(X_{n+1})(\mathbb{1}\{L(\hat{F}(\mathbf{C}_{n+1}; \hat{\tau}_{rand.}(X_{n+1})), \mathbf{W}_{n+1}) \leq \lambda\} - (1 - \alpha))] = 0.$$

As above, we can make this guarantee concrete by choosing $\mathcal{F}$ to be a linear combination of group indicators specifying the topic of the prompt, i.e., $\mathcal{F} = \{(\mathbb{1}\{X \in G\})^\top \beta \mid \beta \in \mathbb{R}^{|\mathcal{G}|}\}$. For any finite set of groups $\mathcal{G}$, this choice of $\mathcal{F}$ yields the high-probability guarantee,

$$\mathbb{P}(L(F(\mathbf{C}_{n+1}; \hat{\tau}_{n+1}), \mathbf{W}_{n+1}) \leq \lambda \mid X_{n+1} \in G) = 1 - \alpha \qquad \text{for all } G \in \mathcal{G}.$$

Like other conformal guarantees, the result of Theorem 3.1 only holds *marginally* over the calibration data, $\{\mathbf{D}_i\}_{i=1}^{n}$. Nevertheless, Gibbs et al. [10] observe that the calibration-conditional validity of their method concentrates around the nominal level as the calibration set size grows.

## 3.2 Level-adaptive conformal prediction

In addition to adapting the filtering cutoff to $X_{n+1}$, it may also be beneficial to adapt the desired error probability $\alpha$ at test time.[3] In this manuscript, we adjust $\alpha$ to ensure sufficient claim retention for each test output. In other scenarios, we might set a stricter $\alpha$ for high-stakes prompts and a more lenient one otherwise. For now, suppose that we are given some desired level function $\alpha(\cdot)$, and our goal is to ensure that $L(\hat{F}(\mathbf{C}_{n+1}), \mathbf{W}_{n+1}) \le \lambda$ with probability $1 - \alpha(X_{n+1})$.

Recall that in the previous section, we use the pinball loss, $\ell_\alpha(\cdot)$ to learn the $1 - \alpha$ quantile of $S(\mathbf{C}_i, \mathbf{W}_i)$. To control the error rate adaptively, here we will use a data-dependent loss for the $i$-th point, $\ell_{\alpha(X_i)}(\cdot)$. Then, similar to the previous section, we define

$$g_S = \underset{g \in \mathcal{F}}{\operatorname{argmin}} \frac{1}{n+1} \sum_{i=1}^{n} \ell_{\alpha(X_i)}(S(\mathbf{C}_i, \mathbf{W}_i) - g(X_i)) + \frac{1}{n+1} \ell_{\alpha(X_{n+1})}(S - g(X_{n+1})), \quad (8)$$

and filter at the cutoff $\hat{\tau}_{\text{l.a.}}(X_{n+1}) := \sup\{S : S \le g_S(X_{n+1})\}$. As in the previous section, it is more convenient to work with a slightly smaller randomized cutoff, $\hat{\tau}_{\text{l.a., rand.}}$. The following theorem shows that our previous guarantee for fixed $\alpha$ extends to this new setting.

**Theorem 3.2.** *Under the assumptions of Theorem 2.1,*

$$\mathbb{E}[f(X_{n+1})(\mathbb{1}\{L(\hat{F}(\mathbf{C}_{n+1}; \hat{\tau}_{\text{l.a., rand.}}(X_{n+1})), \mathbf{W}_{n+1}) \le \lambda\} - (1 - \alpha(X_{n+1})))] = 0, \ \forall f \in \mathcal{F}.$$

Note that we recover the guarantee presented in (1) by defining

$$\Phi(\cdot) = \{\mathbb{1}\{\alpha(\cdot) \in I\} \cdot \mathbb{1}\{\cdot \in G\} : I \in \mathcal{I}, G \in \mathcal{G}\} \quad \text{and} \quad \mathcal{F} = \{\Phi(\cdot)^\top \beta : \beta \in \mathbb{R}^d\}.$$

While this choice of function class elicits an interpretable guarantee, it can be quite large in practice. This can slow down the computation of the conformal cutoff and reduce claim retention. Alternatively, this class might be chosen to exactly satisfy other popular (and more efficient) approximations to multicalibration such as low-degree multicalibration [11]. In Appendix B.2, we briefly explore the consequences of choosing an insufficiently complex function class in a synthetic regression example. A more substantial exposition of these trade-offs can be found in Gibbs et al. [10].

**Estimating $\alpha(\cdot)$**   While the above theory treats $\alpha(\cdot)$ as fixed, in order to ensure that the outputs of our method meet our desired quality criterion (e.g., small interval length or sufficient claim retention) we will need to learn $\alpha(\cdot)$ from the data. To do this, we split our training data into two folds: one is used to estimate $\alpha(\cdot)$ and the other is used to run the calibration method described above. After making these splits, $\alpha(\cdot)$ can be learned using any regression method. In our experiments, we will aim to learn the smallest possible values of $\alpha(\cdot)$ that meet our target quality criterion. A detailed description of our procedure for doing so is given in Appendix B.1.

## 3.3 Conditional boosting

In this section, we describe a new method for automated conformity score design subject to *conditional coverage* constraints. As discussed in the introduction, the goal of this procedure is to find conformity scores that when combined with our filtering method, allow the filter to retain as much of the LLM output as possible while still ensuring validity. For the sake of simplicity, we will assume that we are boosting the conformity score defined in (6). Our approach, however, will generalize to any parameterized score function. We note here that the concurrent work of Kiyani et al. [16] presents another approach to this problem by reframing it as a min-max optimization task.

Let $p_\theta(\cdot)$ denote a *parameterized* claim-scoring function that assigns a measure of confidence to each sub-claim. We run the conditional conformal method on a set of size $n$ and optimize $\theta$ such that the number of retained claims is maximized on a hold-out set of size $m$, i.e.,

$$\theta^* = \underset{\theta}{\operatorname{argmax}} \sum_{i=1}^{m} \sum_{j=1}^{k_{n+i}} \mathbf{1}\left\{p_\theta(P_{n+i}, C_{(n+i)j}) \ge \hat{\tau}_i\right\}. \quad (9)$$

---

[3]Although our methods' implementations differ, we remark here that the level-adaptive guarantee is inspired by a similar result in Zhang and Candès [33].

Here, $\hat{\tau}_i$ denotes the filtering threshold output by the conditional conformal method on the held-out test point $X_{n+i}$. Crucially, $\hat{\tau}_i = \hat{\tau}_i(\theta)$ carries a hidden dependence on $\theta$: $\hat{\tau}_i$ is obtained by solving a quantile regression problem on scores that depend on $p_\theta(\cdot)$.

There are therefore two obstacles to optimizing (9) via gradient descent. First, the indicator function is non-differentiable. To resolve this, we proceed as in Stutz et al. [27] and approximate the indicator using a sigmoid function. Second, it is unclear how to backpropagate through $\hat{\tau}_i(\theta)$.

Observe that for a linear function class $\mathcal{F} = \{\Phi(X)^\top \beta : \beta \in \mathbb{R}^d\}$, the regression (8) by which we obtain $\hat{\tau}_i(\theta)$ is a linear program. Using this observation, we find that $\hat{\tau}_i(\theta)$ can be expressed as the solution to a linear system of equations given by a subset of the dataset that we denote by $B$, i.e.,

$$\hat{\tau}_i(\theta) = \Phi(X_{n+i})^\top \left( \Phi(X)_B^{-1} S_B(\theta) \right). \tag{10}$$

Here, $\Phi(X)_B^{-1}$ and $S_B(\theta)$ are used to denote the matrix of features and vector of scores for the subset of the data given by $B$. In the linear programming literature, this subset is typically referred to as the "optimal basis." It can be explicitly identified by selecting the points at which the quantile regression interpolates the scores. Then, so long as $B$ is invariant to small perturbations of $\theta$ and $S$ is differentiable in $\theta$, we can obtain the derivative of $\hat{\tau}_i(\theta)$ with respect to $\theta$ by backpropagating through (10). Proposition 3.1 identifies a simple condition under which this is possible.

**Proposition 3.1.** *Let $\hat{\tau}_i(\theta)$ denote the non-randomized filtering threshold defined via (8). Assume that the threshold is finite and that the augmented quantile regression for all $S > \hat{\tau}_i(\theta)$ admits a unique non-degenerate basic solution. Let $B$ denote this basis. Then, $\hat{\tau}_i(\theta)$ has derivative*

$$\partial_\theta \hat{\tau}_i(\theta) = \Phi(X_{n+i})^\top \left( \Phi(X)_B^{-1} \partial_\theta S_B(\theta) \right). \tag{11}$$

The technical condition on the basis in Proposition 3.1 nearly always holds; if it does not, we remark that uniqueness can always be obtained by "dithering" the scores and/or design matrix, i.e., adding small i.i.d. noise, at each step of the boosting procedure [7].

The rest of our procedure emulates the **ConfTr** algorithm of Stutz et al. [27]. Having set aside a portion of the dataset for the final calibration procedure, we replicate the conformal prediction algorithm at each iteration. Namely, we randomly split the remaining data into a "calibration" and "test" set. We then run the level-adaptive conformal prediction method on the calibration set, compute the quality criterion on the test set, and backpropagate on the objective given by (9). Algorithm 1, which presents a complete description of the procedure, can be found in Appendix C.

## 4 Experiments

In this section, we present a subset of our experimental results on two previously studied benchmarks in medical question-answering and biography generation. Additional experiments on these data sets, as well as further validation on a synthetic dataset, can be found in Appendix D.

### 4.1 Medical long-form question-answering

In this section, we summarize the experimental setup for the claims and figures presented in Section 1.1. Our experiment considers the long-form medical question-answering dataset (**MedLFQA**) published in Jeong et al. [13]. It combines several previously established benchmarks in the medical question-answering literature: **HealthSearchQA** ($n = 3047$), **K-QA** ($n = 1077$), **LiveQA** ($n = 100$) and **MedicationQA** ($n = 627$). Each prompt in these datasets is also accompanied by either an LLM or human-generated response to each question [6, 1, 19, 26]. Following prior work [19, 13], we treat these responses as ground-truth for model evaluation. Also matching Jeong et al. [13], in our plots, we distinguish between the subset of questions ($n = 201$) in the K-QA dataset that have gold-standard physician responses (denoted by K-QA Golden) and those questions with LLM-generated responses (denoted by K-QA Silver). To reproduce our results, our GitHub repository includes a filtered and combined dataset that removes some non-health-related prompts contaminating the **HealthSearchQA** dataset.

To obtain the dataset used in our experiment, we query GPT-3.5-Turbo with each of the prompts in the combined dataset, and then ask GPT-4o to parse these prompts into self-contained sub-claims. Then, to obtain our "ground-truth" labels, we ask GPT-3.5-Turbo to evaluate whether each claim is

substantiated by the pseudo-ground-truth response for that prompt. The prompts we use for each of these tasks are included in Appendix E.2.

In all of the medical question-answering experiments displayed, we run the conditional conformal method with the following scoring function,

$$S(\mathbf{C}_i, \mathbf{W}_i) = \inf\{\tau : \hat{F}(\mathbf{C}_i; \tau) \text{ contains no unsubstantiated claims}\}.$$

To run our procedure, we must first split the dataset into two parts. We use the first split to both run the conditional boosting method and estimate $\alpha(\cdot)$, while we use the second split to run level-adaptive conformal prediction and evaluate our method's guarantees.

On the first split ($n = 1421$), we boost our claim score and subsequently estimate the level required for 70% claim retention. We achieve the former by optimizing a linear combination of four scores previously described in the LLM factuality literature: the frequency, self-evaluation, and ordinal scores described in Mohri and Hashimoto [21] and the token-level probability assigned to a single-token self-evaluation [14]. Appendix E.1 describes how these claim scores are computed. The function class $\mathcal{F}$ for which we run the conditional boosting method is defined by the linear combination of several prompt-response features. For the sake of brevity, we defer a detailed description of this function class, our method for estimating $\alpha(\cdot)$, and ablations of our method to Appendix D.2.

On the second half of this dataset, we run level-adaptive conformal prediction over the $\mathcal{F}$ used in the conditional boosting step augmented by indicator functions that correspond to $\alpha(\cdot)$ falling in some sub-interval with endpoints lying in $\{0, 0.05, \dots, 1\}$. Finally, to produce the plots in Figure 2, we run this step over 100 random calibration-test splits of size 2354 and 500, respectively.

## 4.2 Wikipedia biographies

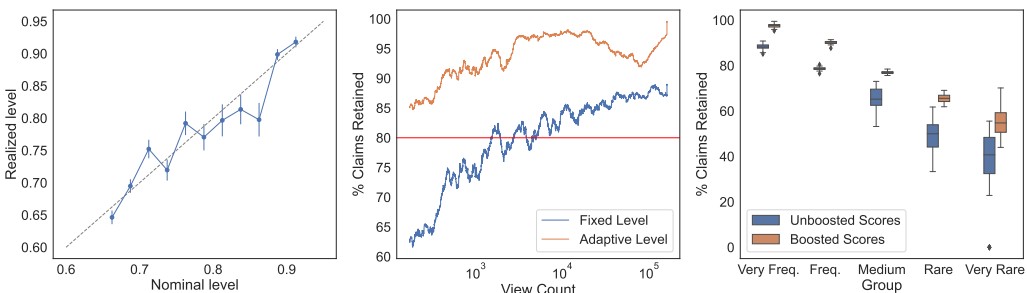

**Figure 3.** Empirical demonstration of our method on the Wikipedia biographies dataset. The left two panels display results for our level-adaptive method, which aims to issue biographies with 3 or fewer errors, while retaining at least 80% of the original claims in the prompt. The left panel compares the binned nominal probabilities reported by our method with bin width 2.5% against the true realized empirical values evaluated over 381 test points and 100 calibration-test splits. The center panel compares the number of claims retained by our method (orange) against the fixed level method of Mohri and Hashimoto [21] (blue). In this plot, the $y$-axis displays a moving average of the number of claims retained with window size 1000, while the $x$-axis a moving average of the number of views (in Jan. 2023) of the Wikipedia article associated with each prompt on the log-scale. Finally, the right-hand panel displays the claim retention obtained with boosted (orange) and unboosted (blue) scores at a fixed level of $\alpha = 0.1$. Boxplots in this panel show the distribution of retentions for 100 calibration-test splits each containing 7246 calibration points and 381 test points.

Our second experiment reconsiders the biography dataset analyzed in Mohri and Hashimoto [21]. We sample 8516 names from Wikipedia and query GPT-3.5-Turbo with the prompt "Write me a short biography of [NAME]." We then ask GPT-4o to parse the prompts into self-contained subclaims. While the prior work only annotates 25 biographies, we need to scale up this experiment to validate our proposed methods. To get around expensive human annotation, we use a variant of the FActscore procedure developed by Min et al. [20]. Their method includes relevant Wikipedia passages identified by the BM25 ranking function in the prompt to the LLM and asks if the claims are supported. Scoring of the LLM outputs (i.e., computation of $p(P_i, C_{ij})$) is done using the frequency scoring method in all displayed figures except for the boosting comparison (the right panel of Figure 3). In that plot, we compare the claim retention achieved by an optimal ensemble of

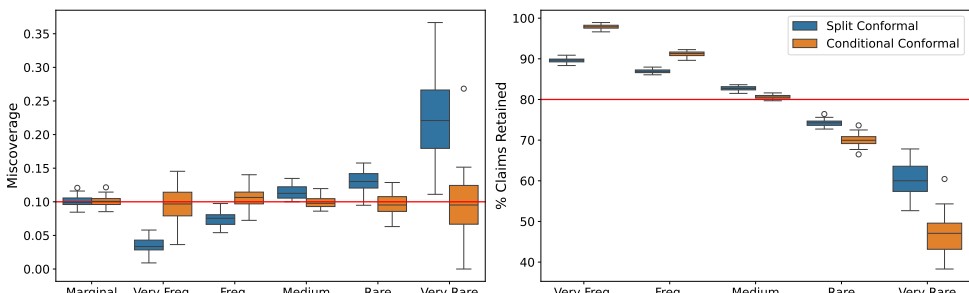

**Figure 4.** Comparison of the split conformal calibration method of Mohri and Hashimoto [21] (blue) against our conditional conformal method (orange). The left and right panels displays the miscoverage and percentage of claims retained by the two methods against the number of views received by the Wikipedia pages in January 2023. The displayed boxplots are computed over 200 trials in which we run both methods on a calibration set of 5890 points and evaluate their coverage on a test set of size 2500. The displayed groups correspond to view counts that are binned into the intervals $(-\infty, \infty)$, $[1000000, \infty)$, $[100000, 1000000)$, $[1000, 100000)$, $[100, 1000)$, $[0, 100)$. The fraction of the data set belonging to each group (in plotted order) is $8.7\%, 30.9\%, 39.5\%, 19.3\%$, and $1.5\%$, respectively.

three cheaper-to-compute scores (self-evaluation, ordinal, log-probability) to the retention of a uniform mixture of those scores [21, 14]. A detailed description of the claim scoring and data collection methods mentioned in this paragraph can be found in Appendices E.1 and E.2. Method ablations and experiments comparing the efficacy of each claim scoring approach are included in Appendix D.3.

In all of the experiments displayed here, we run the conditional conformal method with the scoring function given by

$$S(\mathbf{C}_i, \mathbf{W}_i) = \inf\{\tau : \hat{F}(\mathbf{C}_i; \tau) \text{ contains at most 3 false claims}\}.$$

Empirically, we find that the claim scoring methods we apply are not as well-correlated with factuality as they were for **MedLFQA**. As a result, in order to issue non-trivial guarantees, i.e., to ensure both that $1 - \alpha(X_{n+1})$ remains reasonably large and that the method does not filter too much of the response, we allow the filter to keep up to three false claims.[4] Experimental results for the more typical 0-error guarantee are also included in Appendix D.3. For the sake of avoiding redundancy with the previous subsection, we defer the remaining experimental details to Appendix D.3.

Experimental results displaying the effectiveness of our level-adaptive and conditional boosting procedures on this dataset can be found in Figure 3. In Figure 4, we also provide additional comparisons contrasting the conditional conformal method of Section 3.1 with the marginal method proposed in Mohri and Hashimoto [21]. As anticipated by our theory, we find that our method provides accurate coverage regardless of the popularity of the Wikipedia page, while the split conformal method of Mohri and Hashimoto [21] gives variable results (left panel). As the right panel of the figure shows, this conditional accuracy is obtained by retaining more claims for frequently viewed topics and less claims for rare topics on which the LLM is more uncertain.

## 5   Limitations

While we believe the described methodology can improve LLM factuality, we highlight some limitations here. First, our theoretical results assume that the prompt-response tuples are i.i.d. In settings where user interactions change over time, the test prompts may be incomparable to previous prompts. Even though our framework can be applied to guarantee validity under a pre-specified class of covariate shifts (see Appendix A.2 for details), robustness guarantees under other types of distribution shift will require additional research. Second, since our filter is a wrapper around existing claim scores and thus the utility of our method depends on the quality of the underlying scoring algorithm. If the claim scores are weakly correlated with factuality, the filtered output will be accompanied by a vacuous nominal guarantee. Nevertheless, discovering new features that accurately predict LLM hallucinations is an active field of research, and our wrapper can easily leverage new approaches.

---

[4]While a non-zero error guarantee is not always desirable, it may be suitable for tasks where the cost of an error is low and/or high error rates appear unavoidable, e.g., "Please suggest ten recommendations for my vacation." or "Give me a list of already-approved drugs that might be repurposed to treat COVID."

## Acknowledgments

E.J.C. was supported by the Office of Naval Research grant N00014-24-1-2305, the National Science Foundation grant DMS-2032014, and the Simons Foundation under award 814641. I.G. was also supported by the Office of Naval Research grant N00014-24-1-2305 and the Simons Foundation award 814641, as well as additionally by the Overdeck Fellowship Fund. J.J.C. was supported by the John and Fannie Hertz Foundation. The authors are grateful to Anav Sood and Tim Morrison for helpful discussion on this work.

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

# A  Additional details about conditional conformal

In addition to the methods discussed in Section 2.1, Gibbs et al. [10] also consider extensions of Theorem 2.1 to other function classes and regularized quantile regression procedures. In this section, we show how all of these results can be applied to our present setting. We then describe the randomization procedure that is used to construct our filtering threshold, $\hat{\tau}_{\text{l.a. rand.}}(X_{n+1})$ and prove Theorems 3.1 and 3.2.

Following Gibbs et al. [10], let $\mathcal{F}$ be a vector space and $\mathcal{P} : \mathcal{F} \to \mathbb{R}$ be a convex penalty. Then, to extend our procedure to such function classes and regularizers, we consider regressions of the form

$$g_S = \underset{g \in \mathcal{F}}{\operatorname{argmin}} \frac{1}{n+1} \sum_{i=1}^{n} \ell_{\alpha(X_i)}(S(\mathbf{C}_i, \mathbf{W}_i) - g(X_i)) + \frac{1}{n+1} \ell_{\alpha(X_{n+1})}(S - g(X_{n+1})) + \mathcal{P}(g). \quad (12)$$

Following our work in the main text, this leads to the filtering threshold $\hat{\tau}_{\text{l.a.}}(X_{n+1}) = \max\{S : S \le g_S(X_{n+1})\}$. As discussed earlier, this definition of $\hat{\tau}_{\text{l.a.}}(X_{n+1})$ can be slightly conservative and thus it can be preferrable to work with a randomized analog. We derive this analog now. First, note that (12) is equivalent to the program

$$\underset{p, q \in \mathbb{R}^{n+1}, \, g \in \mathcal{F}}{\text{minimize}} \quad \sum_{i=1}^{n+1} (1 - \alpha(X_i)) p_i + \alpha(X_i) q_i + (n+1)\mathcal{P}(g),$$

$$\text{subject to} \quad S(\mathbf{C}_i, \mathbf{W}_i) - g(X_i) - p_i + q_i = 0, \; \forall 1 \le i \le n,$$
$$S - g(X_{n+1}) - p_{n+1} + q_{n+1} = 0,$$
$$p_i, q_i \ge 0, \; \forall 1 \le i \le n+1.$$

Our randomized variant of $\hat{\tau}(X_{n+1})$ will be based on the dual of this program. Following the calculations of Gibbs et al. [10], the Lagrangian for this problem is

$$\sum_{i=1}^{n+1} (1 - \alpha(X_i)) p_i + \alpha(X_i) q_i + (n+1)\mathcal{P}(g) + \sum_{i=1}^{n} \eta_i (S(\mathbf{C}_i, \mathbf{W}_i) - g(X_i) - p_i + q_i)$$

$$+ \eta_{n+1}(S - g(X_{n+1}) - p_{n+1} + q_{n+1}) - \sum_{i=1}^{n+1} (\lambda_i p_i + \xi_i q_i),$$

where $\lambda_i$ and $\xi_i$ are constrained to be non-negative. Letting $\mathcal{P}^*(\eta) = -\min_{g \in \mathcal{F}}((n+1)\mathcal{P}(g) - \sum_{i=1}^{n+1} \eta_i g(X_i))$ and minimizing the Lagrangian over $g$ gives

$$\sum_{i=1}^{n+1} (1 - \alpha(X_i)) p_i + \alpha(X_i) q_i + \sum_{i=1}^{n} \eta_i S_i(\mathbf{C}_i, \mathbf{W}_i) + \eta_{n+1} S - \mathcal{P}^*(\eta) + \sum_{i=1}^{n+1} \eta_i(q_i - p_i) - \sum_{i=1}^{n+1} (\lambda_i p_i + \xi_i q_i),$$

and additionally minimizing over $p_i$ and $q_i$ produces the constraints

$$\lambda_i = (1 - \alpha(X_i)) - \eta_i, \; \xi_i = \alpha(X_i) + \eta_i.$$

Finally, since $\lambda_i, \xi_i \ge 0$, these constraints are equivalent to the inequalities $-\alpha(X_i) \le \eta_i \le (1 - \alpha(X_i))$. Putting this all together, we arrive at the dual formulation

$$\underset{\eta \in \mathbb{R}^{n+1}}{\text{maximize}} \quad \sum_{i=1}^{n} \eta_i S(\mathbf{C}_i, \mathbf{W}_i) + \eta_{n+1} S - \mathcal{P}^*(\eta),$$

$$\text{subject to} \quad -\alpha(X_i) \le \eta_i \le (1 - \alpha(X_i)), \; \forall 1 \le i \le n+1.$$

Let $\eta^S$ denote a vector of solutions to this program. Then, the critical observation of Gibbs et al. [10] is that the above calculations give a close relationship between the event $S \le g_S(X_{n+1})$ and the value of $\eta_{n+1}^S$. In particular, assuming that strong duality holds, complementary slackness tells us that $S > g_S(X_{n+1}) \implies \eta_{n+1}^S = 1 - \alpha(X_{n+1})$, $S < g_S(X_{n+1}) \implies \eta_{n+1}^S = -\alpha(X_{n+1})$, and $\eta_{n+1}^S \in (-\alpha(X_{n+1}), 1 - \alpha(X_{n+1})) \implies S = g_S(X_{n+1})$. The randomization scheme derived in Gibbs et al. [10] for fixed alpha then corresponds to randomizing over the event that $S = g_S(X_{n+1})$. In particular, following that work, here we define the randomized threshold

$$\hat{\tau}_{\text{l.a. rand.}}(X_{n+1}) = \max\{S : \eta_{n+1}^S \le U\},$$

where $U \sim \text{Unif}([-\alpha(X_i), 1 - \alpha(X_i)])$ is a random variable drawn independent of the data. The following theorem states the coverage guarantee of this method.

**Theorem A.1.** *Assume that $\mathcal{F}$ is a vector space and that for all $f, g \in \mathcal{F}$, the derivative of $\epsilon \mapsto \mathcal{P}(g + \epsilon f)$ exists. Assume additionally that strong duality holds between the primal and dual programs derived above and that $\eta^{S(\mathbf{C}_{n+1}, \mathbf{W}_{n+1})}$ is computed using an algorithm that is symmetric in the input data. Finally, assume that $\{(P_i, R_i, X_i, \mathbf{C}_i, \mathbf{W}_i)\}_{i=1}^{n+1}$ are exchangeable and that $L(\cdot, \cdot)$ is monotone and such that $L(\emptyset, \cdot) = 0$. Then, for all $f \in \mathcal{F}$,*

$$\mathbb{E}[f(X_{n+1})(\mathbb{1}\{L(\hat{F}(\mathbf{C}_{n+1}; \hat{\tau}_{\text{l.a. rand.}}(X_{n+1})), \mathbf{W}_{n+1}) \leq \lambda\} - (1 - \alpha(X_{n+1})))]$$
$$= -\mathbb{E}\left[\frac{d}{d\epsilon}\mathcal{P}(g_{S(\mathbf{C}_{n+1}, \mathbf{W}_{n+1})} + \epsilon f)\Big|_{\epsilon=0}\right].$$

This theorem has two key assumptions that may appear unusual at first glance. First, we have assumed that the dual solutions $\eta^{S(\mathbf{C}_{n+1}, \mathbf{W}_{n+1})}$ are computed using an algorithm that is symmetric in the input data. This is an extremely minor assumption and is satisfied by any standard method for solving convex programs (e.g. interior point solvers). For a much more detailed exposition about efficient algorithms for computing $\{S : \eta_{n+1}^S \leq U\}$, we refer the reader to the original work of Gibbs et al. [10]. Second, we have assumed that strong duality holds. As we will see shortly, this assumption is always satisfied for the linear function classes considered in the main text. For more general vector spaces, Gibbs et al. [10] prove that this condition holds for other common choices of $\mathcal{F}$ such as reproducing kernel Hilbert spaces and Lipschitz functions.

We now prove our main results.

**Proof** [Proof of Theorem A.1] By Theorem 4 of Gibbs et al. [10], $S \mapsto \eta_{n+1}^S$ is non-decreasing in $S$. So, by the monotonicity assumption on $L(\cdot, \cdot)$ and our definition of the conformity score, we have that

$$\mathbb{1}\{L(\hat{F}(\mathbf{C}_{n+1}; \hat{\tau}_{\text{l.a. rand.}}(X_{n+1})), \mathbf{W}_{n+1}) \leq \lambda\} \Leftrightarrow \mathbb{1}\{\eta_{n+1}^{S(\mathbf{C}_{n+1}, \mathbf{W}_{n+1})} \leq U\}.$$

Thus,

$$\mathbb{E}[f(X_{n+1})(\mathbb{1}\{L(\hat{F}(\mathbf{C}_{n+1}; \hat{\tau}_{\text{l.a. rand.}}(X_{n+1})), \mathbf{W}_{n+1}) \leq \lambda\} - (1 - \alpha(X_{n+1})))]$$
$$= \mathbb{E}[f(X_{n+1})(\mathbb{1}\{\eta_{n+1}^{S(\mathbf{C}_{n+1}, \mathbf{W}_{n+1})} \leq U\} - (1 - \alpha(X_{n+1})))]$$
$$= \mathbb{E}[\mathbb{E}_U[f(X_{n+1})(\mathbb{1}\{\eta_{n+1}^{S(\mathbf{C}_{n+1}, \mathbf{W}_{n+1})} \leq U\} - (1 - \alpha(X_{n+1}))) \mid X_{n+1}, \eta_{n+1}^{S(\mathbf{C}_{n+1}, \mathbf{W}_{n+1})}]]$$
$$= -\mathbb{E}[f(X_{n+1})\eta_{n+1}^{S(\mathbf{C}_{n+1}, \mathbf{W}_{n+1})}].$$

For ease of notation let $S_i := S(\mathbf{C}_i, \mathbf{W}_i)$, for $1 \leq i \leq n+1$. Investigating the case where $S = S_{n+1}$ gives us the Lagrangian

$$\sum_{i=1}^{n+1}(1 - \alpha(X_i))p_i + \alpha(X_i)q_i + (n+1)\mathcal{P}(g_{S_{n+1}}) + \sum_{i=1}^{n+1}\eta_i^{S_{n+1}}(S_i - g_{S_{n+1}}(X_i) - p_i + q_i)$$
$$- \sum_{i=1}^{n+1}(\lambda_i p_i + \xi_i q_i).$$

Now, following the calculations in the proof of Proposition 4 of Gibbs et al. [10], we find that taking a derivative along the direction $\epsilon \mapsto g_{S_{n+1}} + \epsilon f$ and applying KKT stationarity to the above gives the equation,

$$0 = \frac{d}{d\epsilon}(n+1)\mathcal{P}(g_{S_{n+1}} + \epsilon f)\Big|_{\epsilon=0} - \sum_{i=1}^{n+1}\eta_i^{S_{n+1}}f(X_i).$$

So, by the exchangeability of the data we have that

$$-\mathbb{E}[f(X_{n+1})\eta_{n+1}^{S(\mathbf{C}_{n+1}, \mathbf{W}_{n+1})}] = -\frac{1}{n+1}\mathbb{E}[\sum_{i=1}^{n+1}f(X_i)\eta_i^{S_{n+1}}]$$
$$= -\mathbb{E}\left[\frac{d}{d\epsilon}(n+1)\mathcal{P}(g_{S_{n+1}} + \epsilon f)\Big|_{\epsilon=0}\right],$$

as desired. $\qquad\square$

## A.1 Proof of Theorem 3.1 and Theorem 3.2

Theorems 3.1 and 3.2 from the main text can now be proven directly as corollaries of Theorem A.1.

**Proof** [Proof of Theorem 3.1] This result is the special case of Theorem A.1 in which $\mathcal{P}(\cdot) = 0$, $\alpha(X_i) = \alpha$ is a constant function, and $\mathcal{F}$ is a finite-dimensional linear class. Under this setup, the primal and dual programs outlined above are linear programs. Since the dual program admits the interior feasible solution $\eta = 0$, Slater's condition immediately implies that these programs satisfy strong duality. □

**Proof** [Proof of Theorem 3.2] This result is the special case of Theorem A.1 in which $\mathcal{P}(\cdot) = 0$ and $\mathcal{F}$ is a finite-dimensional linear class. Similar to the previous result, the resulting primal and dual programs are linear and satisfy strong duality by Slater's condition. □

## A.2 Interpretation of the guarantees in terms of covariate shifts

As discussed in the original work of Gibbs et al. [10], the guarantees on the loss we developed in Theorems 3.1 and 3.2 can be readily interpreted as a form of robustness with respect to a class of distribution shifts affecting $P_X$ (but not $P_{Y|X}$) known as covariate shifts.

First, to precisely define such a shift, suppose that $f \in \mathcal{F}$ is non-negative. Recall that our features $X_{n+1} = X(P_{n+1}, R_{n+1})$ are functions of the prompt and response. Consider the setting in which the training data, $\{(P_i, R_i, \mathbf{C}_i, \mathbf{W}_i)\}_{i=1}^n$ are sampled i.i.d. from a distribution $Q$, while the test point is sampled from the "covariate-shifted" distribution

$$(P_{n+1}, R_{n+1}) \sim \frac{f(X_{n+1})}{\mathbb{E}_Q[f(X)]} dQ_{(P,R)}, \qquad (\mathbf{C}_{n+1}, \mathbf{W}_{n+1}) \sim Q_{(\mathbf{C},\mathbf{W})|(P,R)}.$$

Here, we use the notation $\frac{f(X_{n+1})}{\mathbb{E}_Q[f(X)]} dQ_{(P,R)}$ to refer to the distribution in which the covariate space is re-weighted by $f$, e.g. using the subscript $f$ to denote the re-weighting, we have that for any set $A$,

$$\mathbb{P}_f(X_{n+1} \in A) = \frac{\mathbb{E}_Q[f(X)\mathbf{1}\{X \in A\}]}{\mathbb{E}_Q[f(X)]}.$$

Then, using these definitions, our Theorems 3.1 and 3.2 have the following immediate corollaries.

**Corollary A.1** (Extension of Theorem 3.1). *Let $\mathbb{P}_f$ denote the covariate shift setting described above. Then, under the setting and assumptions of Theorem 3.1,*

$$\mathbb{P}_f(L(\hat{F}(\mathbf{C}_{n+1}; \hat{\tau}_{\text{rand}}(X_{n+1})), \mathbf{W}_{n+1}) \leq \lambda) = 1 - \alpha, \ \forall f \in \{h \in \mathcal{F} : 0 < \mathbb{E}_Q[h(X)] < \infty\}.$$

**Corollary A.2** (Extension of Theorem 3.2). *Let $\mathbb{E}_f[\cdot]$ denote expectations with respect to the covariate shift setting described above. Then, under the setting and assumptions of Theorem 3.2,*

$$\mathbb{E}_f[(\mathbf{1}\{L(\hat{F}(\mathbf{C}_{n+1}; \hat{\tau}_{\text{l.a., rand.}}(X_{n+1})), \mathbf{W}_{n+1}) \leq \lambda\} - (1 - \alpha(X_{n+1})))] = 0,$$
$$\forall f \in \{h \in \mathcal{F} : 0 < \mathbb{E}_Q[h(X)] < \infty\}.$$

# B Details of the level-adaptive procedure

## B.1 Method for learning $\alpha(\cdot)$

To make our approach to learning $\alpha(\cdot)$ precise, we formally define our quality criterion using a binary function $Q(\cdot, \cdot)$ that takes in a data point and a cutoff and returns 1 or 0 if the criterion is satisfied or not, respectively. For example, for the claim retention objective described in Section 1.1,

$$Q(\mathbf{C}, \hat{\tau}) := \mathbf{1}\left\{\frac{|F(\mathbf{C}; \hat{\tau})|}{|\mathbf{C}|} \geq 0.7\right\}.$$

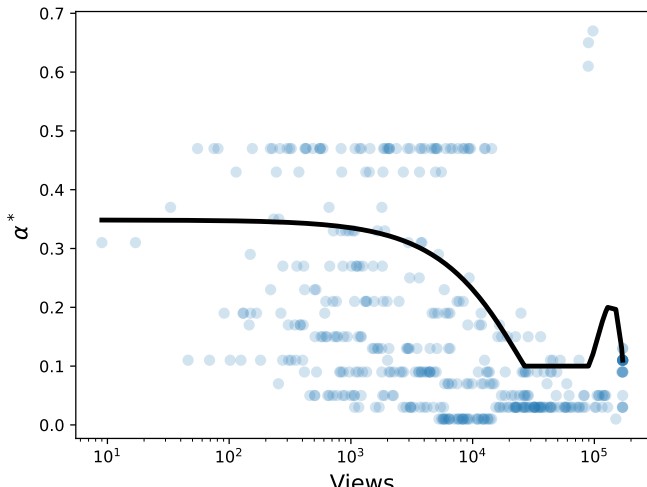

**Figure 5.** For a hold-out set of size $424$, we plot the optimal level threshold, $\alpha_i^*$ for claim retention (13) against the number of Wikipedia page views (on the log scale) for the associated person. Letting $v_i$ denote the view count for person $i$, the black line denotes the estimate of the 0.25-quantile of $\alpha_i^* \mid v_i$ obtained by regressing over the function class given by $\{\beta_0 + \sum_{i=1}^{3} \beta_i v^i \mid \beta \in \mathbb{R}^4\}$.

As discussed in the main text, when running our method we split our training data into a portion for fitting $\alpha(\cdot)$ and portion for calibrating the conformity score. Then, we further divide the portion of the data that is used to fit $\alpha(\cdot)$ into two folds. We use the first fold to run the conditional conformal procedure at a fixed grid of quantiles, $\alpha \in \{0.01, \ldots, 0.99\}$, and the second dataset to evaluate $Q(\mathbf{C}_i; \hat{\tau}_\alpha(X_i))$ for each choice of $\alpha$. Here, $\hat{\tau}_\alpha(X_i)$ denotes the conditional conformal cutoff evaluated for test claim $\mathbf{C}_i$ at fixed level $1 - \alpha$.

After collecting these data, we compute the maximum $\alpha$ at which the quality criterion is achieved. Formally, for each point $(\mathbf{C}_i, X_i)$ in the dataset held out from fitting, we compute

$$\alpha_i^* = \inf\{\alpha : \forall \beta \geq \alpha, \ Q(\mathbf{C}_i, \hat{\tau}_\beta(X_i)) = 1\}. \tag{13}$$

With this data in hand, our goal now is to fit a function $\alpha(\cdot)$ that predicts $\alpha_i^*$ from $(X_i, \alpha_i^*)$. In principle, this could be done using any number of regression algorithms. In our experiments, we choose to estimate a quantile of $\alpha_i^* \mid X_i$ using the same function class we used to run our conditional conformal procedure. We then truncate these estimates to lie above 0.1. See Figure 5 for a plot of this estimated quantile and the underlying data points used to fit this regression. Our choices of quantile and truncation are motivated by our desire to ensure that the adaptive level will guarantee the quality criterion for most test points without issuing completely vacuous guarantees, i.e., fitting some $\alpha(\cdot)$ whose range is concentrated around 1 or 0.

### B.2 Dependence of the level-adaptive guarantee on the function class

The quality of our level-adaptive guarantee strongly depends on the choice of function class, $\mathcal{F}$. To understand this, recall that split conformal prediction corresponds to taking $\mathcal{F}$ to be the class of constant functions, $\mathcal{F} = \{x \mapsto \beta : \beta \in \mathbb{R}\}$. In this case, Theorem 3.2 states that

$$\mathbb{E}[\mathbb{1}\{L(\mathbf{C}_{n+1}, \mathbf{W}_{n+1}) \leq \lambda\}] = \mathbb{E}[(1 - \alpha(X_{n+1}))],$$

i.e. with probability $\mathbb{E}[1 - \alpha(X_{n+1})]$ the error rate of our method lies at the target level. This guarantee is extremely weak; our objective is to ensure that the nominal probability is pointwise close to the true level, i.e. $|\mathbb{P}(L(\mathbf{C}_{n+1}, \mathbf{W}_{n+1}) \leq \lambda \mid \alpha(X_{n+1})) - (1 - \alpha(X_{n+1}))|$ is small. In the main text, we saw empirical results in which our methods were able to approximately achieve this target. These result were obtained by designing $\mathcal{F}$ to include functions that depend on $\alpha(X_{n+1})$.

To demonstrate the importance of this choice of $\mathcal{F}$ more explicitly, Figure 6 compares the realized difference between the true coverage, $\mathbb{P}(Y_{n+1} \in \hat{C}(X_{n+1}) \mid \alpha(X_{n+1}))$ and the nominal level

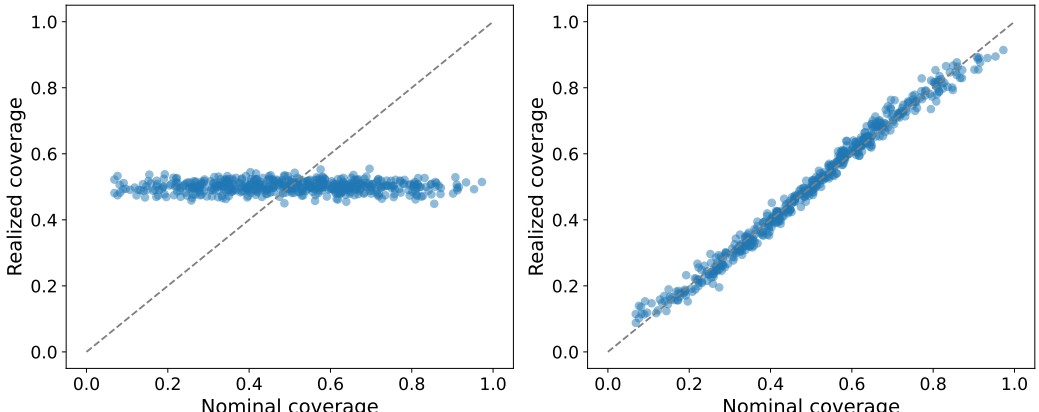

**Figure 6.** Comparison of the realized and nominal levels of our level-adaptive method for various choices of $\mathcal{F}$. Here, $(X_i, Y_i) \sim \mathcal{N}(0, I_2)$ and we set the conformity score to be simply $S(X_i, Y_i) = Y_i$. In this experiment, we use our level-adaptive method (Section 3.2 in the main text) to construct prediction sets for $Y_i$ given covariates $X_i$. We set $\alpha(X) = \sigma(X)$ where $\sigma(\cdot)$ denotes the sigmoid function with temperature 1. We then run the level-adaptive method on a calibration set of size $n = 1000$ and evaluate the method on 1 test point; the plotted points (center, right) are obtained from 500 trials. The left panel shows results for $\mathcal{F} = \{x \mapsto \beta : \beta \in \mathbb{R}\}$, while the right panel displays results for $\mathcal{F} = \{(1, \alpha(X))^\top \beta : \beta \in \mathbb{R}^2\}$.

$1 - \alpha(X_{n+1})$ when $\mathcal{F} = \{x \mapsto \beta : \beta \in \mathbb{R}\}$ and $\mathcal{F} = \{(1, \alpha(X))^\top \beta : \beta \in \mathbb{R}^2\}$ on a simple synthetic dataset. As anticipated, when $\mathcal{F}$ consists only of constant functions, the nominal guarantee is spurious, and the true probability of coverage is uncorrelated with the claimed level (left panel). This is corrected by adding $\alpha(X)$ to the feature space of $\mathcal{F}$; despite being quite simple this choice empirically achieves the desired calibration (right panel). We leave a more substantial investigation of the trade-off between $\mathcal{F}$ complexity, empirical calibration, and efficiency to future work.

## C   Details of the Boosting Procedure

For the boosting procedure, we will assume that we are working with the derandomized variant of the conditional conformal method of Gibbs et al. [10]. In addition, these results will hold only for the finite-dimensional function class variant of their method with no regression penalty, i.e., in the notation of Appendix A we assume that

$$\mathcal{F} = \{\Phi(\cdot)^\top \beta : \beta \in \mathbb{R}^d\}, \qquad \mathcal{P}(\cdot) = 0.$$

We first restate and then prove Proposition 3.1.

**Proposition C.1.** *Let $\hat{\tau}_i(\theta)$ denote the non-randomized filtering threshold defined via* (8). *Assume that the threshold is finite and that the augmented quantile regression for all $S > \hat{\tau}_i(\theta)$ admits a unique non-degenerate basic solution. Let $B$ denote this basis. Then, $\hat{\tau}_i(\theta)$ has derivative*

$$\partial_\theta \hat{\tau}_i(\theta) = \Phi(X_{n+i})^\top \left( \Phi(X)_B^{-1} \partial_\theta S_B(\theta) \right). \tag{14}$$

**Proof**   Assume without loss of generality that $i = 1$. We drop the subscript on $\hat{\tau}$ for notational convenience. To differentiate between the imputed value of the $(n + 1)$-st score and the vector of all observed scores, we will use $S$ to refer to the former and $\mathbf{S}$ to the latter. When the optimal basis is unique, we will assume that the primal solution is defined via the interpolator of the basis. Otherwise, we assume it is obtained via some symmetric algorithm.

First, we establish that $\theta \mapsto \Phi(X_{n+1})^\top \beta_S(\theta)$ is locally linear in $\theta$ for $S > \hat{\tau}(\theta)$. First, observe that the non-interpolated points of the quantile regression correspond to the non-basic variables in the solution, and that their residuals are equivalent to the non-basic "reduced costs" of the LP. Standard results from LP sensitivity analysis establish that when these reduced costs are bounded away from 0 (implied by our assumptions of non-degeneracy and uniqueness), the basis of the LP is unchanged under sufficiently small perturbations of the objective coefficients (i.e., for perturbations of $\theta$) [8].

Since $\beta_S$ is defined by the solution to a linear system of equations involving the basis, the local constancy of the basis implies the claim of local linearity. Second, we establish that $S \mapsto \beta_S(\theta)$ for $S > \hat{\tau}(\theta)$ is constant. This is immediate from the KKT conditions for (7). The optimal basis is unchanged for all $S > \hat{\tau}(\theta)$.

Fix any $\theta_0$. We claim that there exists a ball around $\theta_0$ such that $\hat{\tau}(\theta) = \Phi(X_{n+1})^\top \beta_{\hat{\tau}(\theta_0)+1}(\theta)$.

To prove this, first note that we know that for $\theta$ sufficiently close to $\theta_0$, $\hat{\tau}(\theta_0) + 1 > \Phi(X_{n+1})^\top \hat{\beta}_{\hat{\tau}(\theta_0)+1}(\theta)$. Since $S \mapsto \beta_S(\theta)$ is constant above $\hat{\tau}(\theta)$, this immediately implies that for $S > \Phi(X_{n+1})^\top \hat{\beta}_{\hat{\tau}(\theta_0)+1}(\theta)$, $\hat{\beta}_S(\theta) = \hat{\beta}_{\hat{\tau}(\theta_0)+1}(\theta)$ and thus $S > \Phi(X_{n+1})^\top \hat{\beta}_S(\theta)$. Hence, $\hat{\tau}(\theta) \le \Phi(X_{n+1})^\top \hat{\beta}_{\tau(\theta_0)+1}(\theta)$.

Now, assume by contradiction that $\hat{\tau}(\theta) < \Phi(X_{n+1})^\top \hat{\beta}_{\tau(\theta_0)+1}(\theta)$. For ease of notation let $v = \Phi(X_{n+1})^\top \hat{\beta}_{\tau(\theta_0)+1}(\theta)$. Now, since $\hat{\tau}(\theta) < v$ we must have $v > \Phi(X_{n+1})^\top \hat{\beta}_v(\theta)$. Recall that we additionally know that $\hat{\tau}(\theta_0) + 1 > \Phi(X_{n+1})^\top \hat{\beta}_{\hat{\tau}(\theta_0)+1}(\theta)$. So, by our assumption that $S \mapsto \beta_S(\theta)$ is constant above $\hat{\tau}(\theta)$ we find that $\hat{\beta}_{\tau(\theta_0)+1}(\theta) = \hat{\beta}_v(\theta)$. Thus, $v = \Phi(X_{n+1})^\top \hat{\beta}_{\tau(\theta_0)+1}(\theta) = \Phi(X_{n+1})^\top \hat{\beta}_v(\theta)$, which gives the desired contradiction.

The above proves that $\tau(\theta) = \Phi(X_{n+1})^\top \beta_{\tau(\theta_0)+1}(\theta)$ locally in a ball around $\theta_0$. The desired result follows by our previous proof of the differentiability of the right hand-side.

$\square$

With this result in hand, our boosting procedure is now given in detail in Algorithm 1 below. In this algorithm we use the notation $\Phi(I) = [\Phi(X_i)]_{i \in I}$ and $S_\theta(I) = (S_\theta(\mathbf{C}_i, \mathbf{W}_i))_{i \in I}$ to refer the feature matrix and scores for the data points in $I$.

---

**Algorithm 1:** *Conditional boosting*

---

**Data:** Boosting dataset $\mathcal{D}_{\text{boost}} = \{\mathbf{D}_i\}_{i=1}^m$, conformity score function $S_\theta(\cdot, \cdot)$, function class
    basis $\boldsymbol{\Phi}$, initialization $\theta_0$, sigmoid temperature $\lambda$, level $\alpha$, boosting iteration count $T$.

**for** $t \in [T]$ **do**
    $\ell = 0$;
    $\mathcal{D}_1, \mathcal{D}_2 = \mathsf{RandomSplit}(\mathcal{D}_{\text{boost}})$ ;
    **for** $i \in \mathcal{D}_2$ **do**
        $B = \mathsf{getOptimalBasis}(\alpha, \Phi(\mathcal{D}_1), S_\theta(\mathcal{D}_1), \Phi(\{i\}), S_{\theta_{t-1}}(\{i\}))$;
        $\hat{\beta}_i = \Phi_B^{-1}(\mathcal{D}_1 \cup \{i\}) S_B(\mathcal{D}_1 \cup \{i\})$;
        $\ell = \ell + \sum_{j=1}^{k_i} \sigma_\lambda(\Phi(\{i\})^\top \hat{\beta}_i - p_\theta(P_i, C_{ij}))$;
    $\theta_t = \mathsf{Adam}(\ell, \theta_{t-1}).\mathsf{step}()$;
**return** $\theta_T$.

---

In Algorithm 1 above, $\mathsf{getOptimalBasis}(\cdot)$ refers to the subroutine that computes the optimal basis $B$ appearing in the statement of Proposition C.1 for text point $i$.

In order to decrease computational complexity, the experiments in this paper are run using a simplified version of Algorithm 1. In particular, the subroutine $\mathsf{getOptimalBasis}$ obtains the optimal basis in Algorithm 1 by identifying the $\dim(\Phi)$ points that are interpolated by the regression when $S > \hat{\tau}_i(\theta)$. Identifying this set is computationally burdensome since each gradient step in $\theta$ requires computing $\hat{\tau}_i(\theta)$ (and therefore $\hat{\beta}_i$) for each point in the second split. Thus, the gradient computation step scales quadratically in $|\mathcal{D}_2|$. A linear-time approximation to the gradient can be obtained by defining $B$ as the optimal basis from the quantile regression fit to $\mathcal{D}_1$ alone. This approximation, which does not account for the test-point augmentation in the conditional conformal procedure, is much faster since it yields a single $\hat{\beta}$ to use for all the test points. The experiments in this paper are run using this simplified procedure. To concretely modify Algorithm 1, $B$ is defined as the optimal

basis of the quantile regression fit to $\mathcal{D}_1$ alone, the computation in the nested for loop is removed, and the estimated threshold in the sigmoid function becomes $\Phi^\top(\{i\})\hat{\beta}$.

# D  Additional experiments

## D.1  Synthetic data

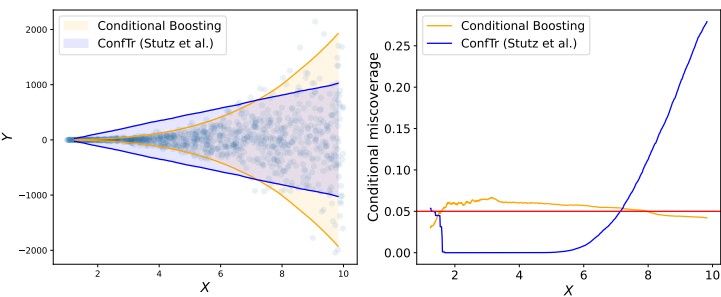

**Figure 7.** Comparison of the marginal boosting procedure of Stutz et al. [27] (blue) against our conditional boosting method (orange). The left panel shows the prediction sets produced by each method, while the right panel displays the conditional coverage, $\mathbb{P}(Y_{n+1} \in \hat{C}(X_{n+1}) \mid X_{n+1}^{(1)})$ against the values of the first feature, $X_{n+1}^{(1)}$. Using the Adam optimizer with learning rate set to 0.001, the plotted scores are boosted for 500 steps on a synthetic dataset of size $n = 1000$ and evaluated on another test dataset of size $n = 2000$.

To demonstrate how our methods improve upon previous approaches, we consider a synthetic regression dataset with substantial heteroskedasticity. The data in this experiment are generated by sampling two features $X_i^{(1)} \overset{\text{iid}}{\sim} \text{Unif}(1, 10)$ and $X_i^{(2)} \overset{\text{iid}}{\sim} \text{Unif}(5, 10)$, while the labels are sampled from a distribution whose variance is given by the sixth power of the first feature, i.e., $Y_i \sim \mathcal{N}(0, (X_i^{(1)})^6)$. We then construct prediction intervals using the score function $S_\theta(X, Y) := |Y|/|X^\top\theta|$.

Figure 7 compares the **ConfTr** algorithm of Stutz et al. [27] against our **Conditional Boosting** method. While the left panel shows that the former method may sometimes issue shorter intervals, these come at the cost of degraded conditional validity (right panel). In particular, the prediction sets resulting from **ConfTr** have extremely poor coverage on the subset of data with high conditional variance. By contrast, learning $\theta$ using our procedure still attains near-perfect conditional coverage.

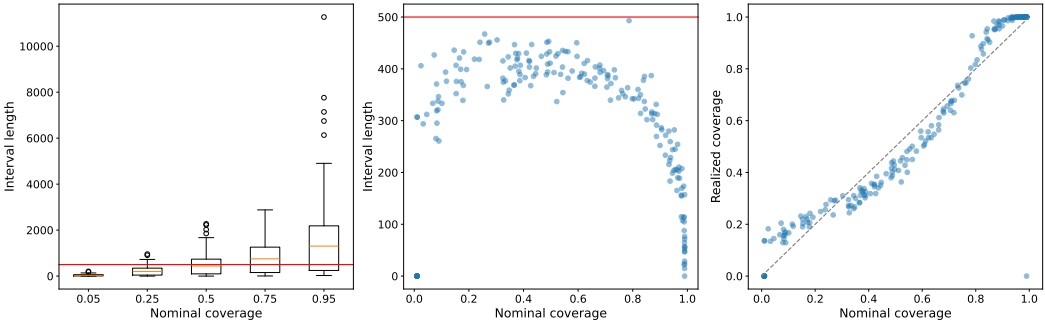

**Figure 8.** Performance of our level-adaptive method on a synthetic dataset. We use $n = 2000$ points to estimate the adaptive $\alpha(\cdot)$ function. We then run the level-adaptive method on a calibration set of size $n = 1000$ and evaluate the method on 1 test point; the plotted points (center, right) are obtained from 200 trials. The left panel shows the distribution of interval lengths obtained on the test set for fixed values of $\alpha \in \{0.05, 0.25, 0.5, 0.75, 0.95\}$. The center panel displays the interval lengths obtained when the level $\alpha(X_{n+1})$ is now chosen adaptively to ensure a maximum prediction set size of at most 500 (red line). Finally, the right panel compares the realized coverage $\mathbb{P}(Y_{n+1} \in \hat{C}(X_{n+1}) \mid \alpha(X_{n+1}))$ against the nominal level, $1 - \alpha(X_{n+1})$ reported by our method with $\mathcal{F} = \{\beta_0 + \sum_{i=1}^{2} \beta_1 x^i + \beta_3 \alpha(x) \mid \beta \in \mathbb{R}^4\}$.

Figure 8 applies the level-adaptive conformal prediction procedure to the same data-generating process. In this example, we use the naive absolute value score $S(X, Y) = |Y|$. Since the score is poorly chosen and the data is noisy, the resulting prediction interval can be very large. The left panel of the figure shows boxplots that display the empirical distribution of interval lengths over the test set at various fixed levels of $\alpha \in \{0.05, 0.25, 0.5, 0.75, 0.95\}$. Unsurprisingly, at most fixed levels for $\alpha$, many intervals are substantially larger than the maximum target length of 500. This is corrected when we fit using an adaptive level, $\alpha(X_i)$ (center panel), which is estimated to ensure that the interval lengths fall below 500. Finally, the right-hand panel verifies that the guarantee of Theorem 3.2 is accurate. The issued values of $\alpha(X_{n+1})$ closely track the true coverage probabilities; note that the latter can be computed in closed-form for our data-generating process.

## D.2 Additional results for Section 4.1

**Additional details for the experiment set-up**    The function class $\mathcal{F}$ is defined by the linear combination of an intercept, the number of characters in the prompt, the number of characters in the response, the mean frequency score assigned to the claims, the standard deviation of the frequency scores assigned to the claims, and group-indicators corresponding to the source dataset. The results shown in Figure 2 are obtained by running the conditional boosting algorithm for 1000 steps using the Adam optimizer with learning rate set to 0.001.

We estimate $\alpha(\cdot)$ by dividing the first split of the data into two further sub-parts. On the first half of this further subdivisionx ($n = 720$), we run the fixed-level conditional conformal procedure (with the same $\mathcal{F}$ used to boost the scores) for each $\alpha$ in a grid of 50 evenly spaced values between 0 and 1. Using the second half of this dataset, we evaluate the percentage of retained claims at each $\alpha$. We record the smallest $\alpha$ for which at least 70% of claims are preserved at *all larger* values of $\alpha$. We then obtain our estimate of $\alpha(\cdot)$ by fitting a 0.85-quantile regression over $\mathcal{F}$ to this dataset. For interpretability and numerical stability in the next step, the final output of the function is truncated to lie between 0.1 and 0.5.

**Additional experiments**    Figure 9 decomposes the effect of each method in this paper on claim retention and calibration for the **MedLFQA** benchmark. For a fixed-level guarantee, we observe that the boosting method substantially improves claim retention. When we adapt the level to guarantee high claim retention, the improvement of the boosting method is visible in the second panel of Figure 9. Namely, we find that when the level-adaptive method is augmented with boosted scores we obtain the same claim retention while issuing stronger guarantees as compared to the un-boosted scores. Finally, the third panel of Figure 9 verifies that our reported values of $1 - \alpha(X_{n+1})$ (approximately) match the realized error rates.

Using the same experimental set-up as the ablations, Figure 10 compares the effect of the choice of claim scoring method on the nominal guarantee offered by the level-adaptive procedure. Note that the log-probability and frequency claim scores appear to offer the strongest set of guarantees. However, the markedly lower level of claim retention for the log-probability score make the guarantees difficult to directly compare.

## D.3 Additional results for Section 4.2

**Additional details for the experiment set-up**    To define features for the conditional conformal function class, we obtain page metadata from the Wikidata API. The metadata we use is the number of views of the biography's Wikipedia article in January 2023. Since this data is quite right-skewed, the view counts for the most popular articles can cause numerical issues in our optimizer. Consequently, we trim the largest entries to the 0.95-quantile of the view data.

Here, we also provide some additional details regarding the function classes used in our biography experiments. In all examples, we aim to provide uniform coverage guarantees regardless of the underlying popularity of the topic. In the fixed-level filtered biographies, we run the conditional conformal method with $\mathcal{F} = \{\beta_0 + \sum_{i=1}^{3} \beta_i v^i \mid \beta \in \mathbb{R}^4\}$, where $v$ denotes the number of views of the corresponding Wikipedia page in January 2023. In the level-adaptive examples, we use a nearly identical workflow to the previous section. The only difference is that here we estimate $\alpha(\cdot)$ using a 0.75-quantile regression over $\mathcal{F}$ (see Figure 5 for a plot of this estimate). Additionally, in the final level-adaptive fit, we add a linear term to this function class given by $\beta_4(\alpha(\cdot) - 0.1)^2$. Note that in

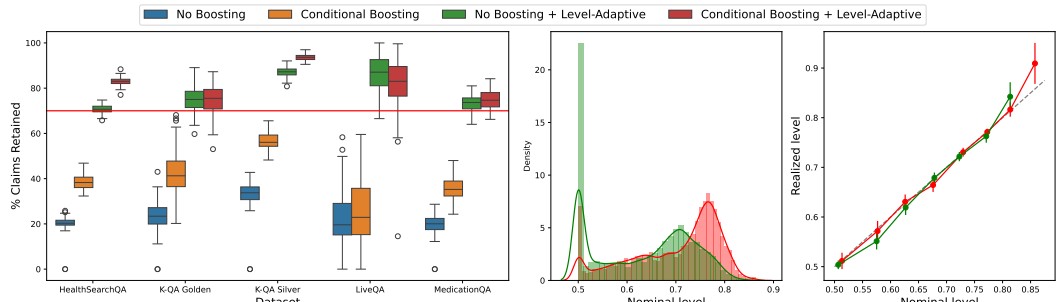

**Figure 9.** The left panel of the figure shows the percentage of claims retained by various methods on the **MedLFQA** benchmark of Jeong et al. (2024). This benchmark contains many datasets each of which is displayed on the x-axis. Before boosting, we define our claim score by an equally-weighted ensemble of previously published claim scoring functions. We run 100 trials in which we resample a boosting/$\alpha(\cdot)$-estimation split ($n = 1441$), calibration split ($n = 2354$), and test split ($n = 500$). We plot 4 filtering methods: the fixed-level ($1 - \alpha = 0.9$) method that guarantees conditional validity over the function class given by dataset indicators and prompt metadata (prompt length, response length, as well as the mean and standard deviation of the "log probability" claim score over the output) (blue), the same conditionally valid method with adaptive level $\alpha(X_{n+1})$ (orange), the conditionally valid method with boosted scores (green), and a combination method that incorporates both conditional-boosting and level-adaptive CP (red). All methods are set-up to ensure that the final output contains **0** false claims. The middle panel shows that boosting allows the level-adaptive procedure to issue guarantees with higher confidence. The right panel verifies that the reported nominal levels $1 - \alpha(X_{n+1})$ match the empirical error frequencies over equi-spaced bins of width 0.05; these bin indicators are included in the function class used in the level-adaptive procedure.

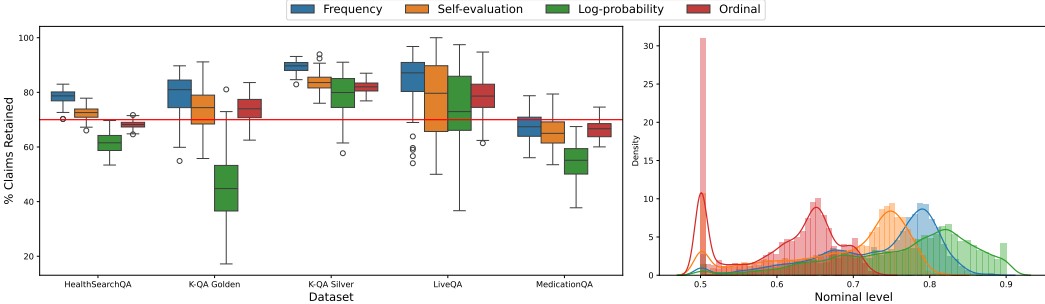

**Figure 10.** The set-up is identical to that of Figure 9. Here we do not consider boosted scores, but display the performance of the level-adaptive method for each claim scoring approach in isolation.

our experiments, we truncate $\alpha(\cdot)$ to lie above 0.1. As a result, a substantial proportion of outputs receive that estimated quantile and thus, $(\alpha(\cdot) - 0.1)^2$ correlates with the more variable portion of the quantile function.

Though the first two panels of Figure 3 and Figure 4 are run using the more performant frequency score, the right panel of Figure 3 is obtained by finding an optimal linear combination of the three cheaper-to-obtain claim scores (self evaluation, token probability, ordinal). We obtain the displayed result by running 1000 steps of Adam (with PyTorch defaults, i.e., learning rate 0.001) through the conditional conformal procedure given by the same $\mathcal{F}$ described in the previous paragraph.

**Additional experiments** Figures 11 and 12 decompose the effect of each method on claim retention and calibration for the **FActscore** benchmark. Notably, we observe that optimally ensembling the 3 cheaper-to-compute claim scores using conditional boosting improves claim retention and/or shifts the distribution of issued guarantees to the right. Furthermore, regardless of the guarantee issued, the third panel of both of these figures confirms that our method is well-calibrated. Finally, as the second panel of Figure 12 confirms, if we aim to guarantee 0 false claims in the final output, it is unlikely that we issue a nominal probability above $50\%$. This motivates our choice to instead guarantee that the filtered output contains at most three false claims.

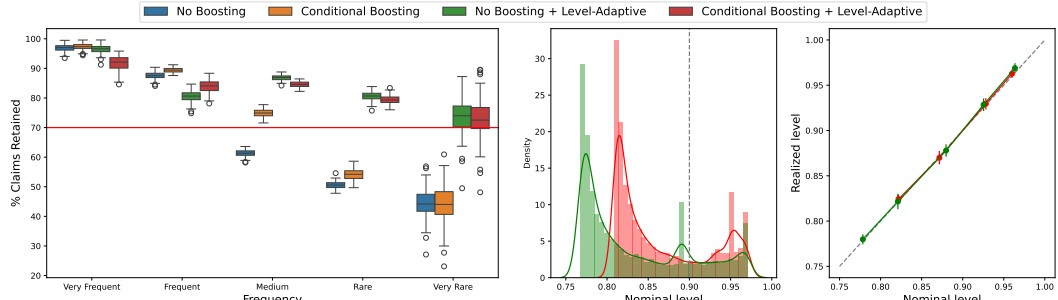

**Figure 11.** This figure replicates the previous ablation study (namely Figure 9) for the **FActscore** dataset of Min et al. [20]. For the sake of brevity, we only describe the differences compared to Figure 9. The frequency of each biography topic varies; the displayed groups correspond to Wikipedia view counts in Jan. 2023 that are binned into the intervals $[1000000, \infty), [100000, 1000000), [1000, 100000), [100, 1000), [0, 100)$. The function class used for calibration is defined by the linear combination of these bin indicators and the view count of each Wikipedia article. We run 100 trials in which we resample a boosting/$\alpha(\cdot)$-estimation split ($n = 847$), calibration split ($n = 5338$), and test split ($n = 500$). All methods are set-up to ensure that (with high probability) the final output contains no more than 3 false claims.

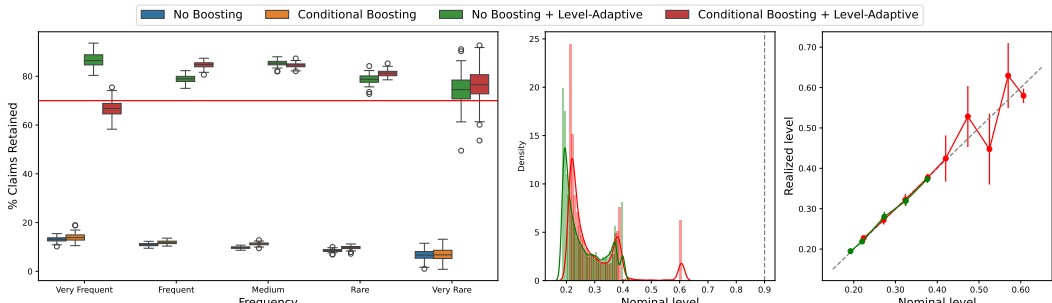

**Figure 12.** The set-up is identical to that of Figure 11, but now all methods are set-up to ensure that (with high probability) the final output contains no more than 0 false claims.

Using the same experimental set-up as the ablations, Figures 13 and 14 compare the effect of the choice of claim scoring method on the nominal guarantee offered by the level-adaptive procedure. Note that once again the log-probability and frequency claim scores appear to offer the strongest set of guarantees.

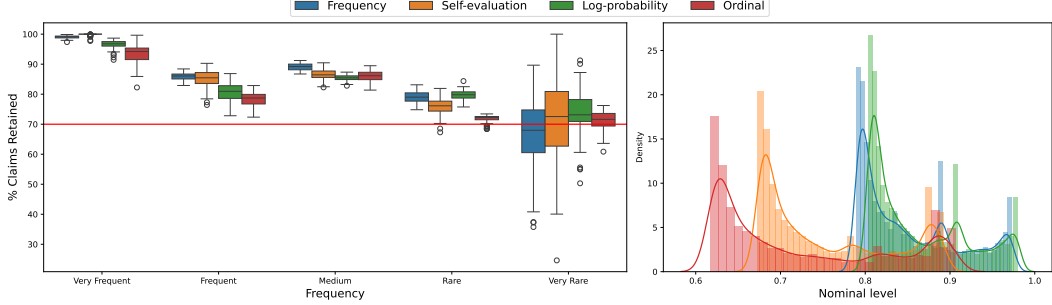

**Figure 13.** The set-up is identical to that of Figure 11. Here we do not consider boosted scores, but display the performance of the level-adaptive method for each claim scoring approach in isolation.

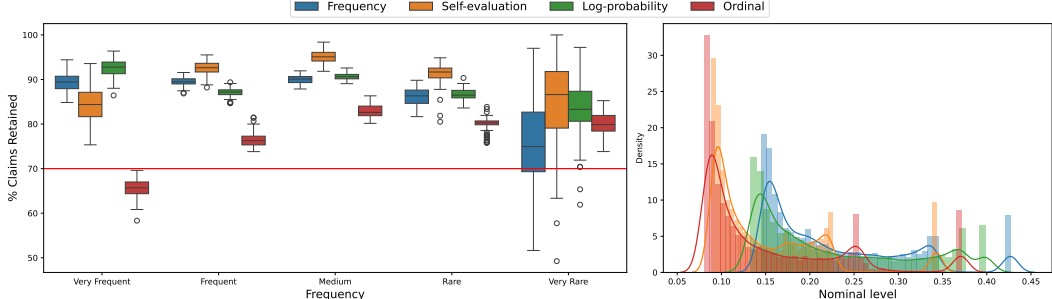

**Figure 14.** The set-up is identical to that of Figure 12. Here we do not consider boosted scores, but display the performance of the level-adaptive method for each claim scoring approach in isolation.

# E    Additional scoring details

## E.1    Claim scores

We collect several features to use for estimating a confidence level in each claim. Unless otherwise noted, the claim scoring function used in this paper is defined by the "frequency" score examined by Mohri and Hashimoto [21]. We compute this score by querying GPT-3.5-Turbo 5 times at a temperature of $T = 5$. For each claim, we re-query the LLM to ask if the claim is supported in the resampled generation. These scores which can be $-1$ for contradicted, $0$ for not present, and $1$ for supported are then averaged over the $5$ responses. For example, a claim that is present in three out of the five responses, but absent in the other two would receive a score of $0.6$.

Since the frequency score requires many API queries, we also investigate several cheaper and faster-to-compute methods for claim scoring. Following Mohri and Hashimoto [21], we also collect the self-reported probability obtained by directly asking the LLM to report its estimated probability of claim correctness to three significant figures. We also compute the ordinal score, i.e., the order in which the claim appeared in the original response. Going beyond Mohri and Hashimoto [21], we compute the LLM's internal probability of each claim by querying the model with the list of claims and asking it to output a single-character (T or F) assessment of its beliefs. The internal probability of the model can then be computed by exponentiating the log probability of the returned token.

## E.2    Prompts

To parse the initial response into scorable sub-claims, we prompt the LLM (GPT-4o, in our experiments). After first parsing the initial response into its component sentences, we copy the method of Min et al. [20] (made available on GitHub under an MIT license) and use in-context learning to improve the parsing accuracy of the LLM. We first use the BM25 ranking function to match our sentences to similar examples previously parsed by Min et al. [20]. We prepend these gold-standard examples as well as the new sentence to be parsed to the prompt,

> Please breakdown the following sentence into independent facts: [...]

To annotate the subclaims, we either retrieve the response given in the MedLFQA dataset or emulate Min et al. [20] and retrieve Wikipedia passages that are most relevant (using the BM25 ranking function) to the subject of the biography. The prompt includes this passage followed by the previous claims in the output that have already been annotated to ensure self-consistency,

> Answer the question about [...] based on the given context and your previous answers. Title: [...] Text: [...] Previous input: [...] True or False? Output: [...] Input: [...] True or False? Output:

To evaluate the support of the claims in new response, we prompt the LLM using

> You will get a list of claims and piece of text. For each claim, score whether the text supports, contradicts, or is unrelated to the claim. Directly return a jsonl, where each line is "id":[CLAIM_ID], "score":[SCORE]. Directly return the jsonl

with NO explanation or ANY other formatting. For the [SCORE], return 1 for supports, -1 for contradicts, and 0 for unrelated. The claims are: [...] The text is: [...]'

We obtain self-assessed probabilities by prompting the LLM using

You will get a list of claims and the original prompt that motivated these claims. For each claim, assess the probability of correctness. Directly return a jsonl, where each line is "id":[CLAIM_ID], "gpt-score":[SCORE]. Directly return the jsonl with NO explanation or ANY other formatting. For the [SCORE], return the esimated probability of correctness to three significant figures. The original promptis: [...] The claims are: [...]

We obtain the internal probability of the model using a very similar prompt,

You will get a list of claims and the original prompt that motivated these claims. For each claim, assess the correctness. Directly return a jsonl, where each line is "id":[CLAIM_ID], "gpt-bool":[BOOL]. Directly return the jsonl with NO explanation or ANY other formatting. For the [BOOL], return "T" or "F" in quotes so that it is valid json. The original prompt is: [...] The claims are: [...]

