# OpenReview forum: "Large language model validity via enhanced conformal prediction methods"
_NeurIPS.cc/2024/Conference — NeurIPS 2024 poster_

### Official Review · Reviewer_tBPp · 2024-07-12

**Soundness:** 3
**Presentation:** 3
**Contribution:** 2
**Rating:** 5
**Confidence:** 5

**Summary:**

The paper proposes a new way to filter out LLM generations such that the resulting text only contains a set number of invalid claims with a certain probability. To do so, the authors adapt a new conformal mechanism to be level adaptive i.e. conditioned on the prompts rarity for example. They also propose a boosted version which additionally reduces the number of wrongly removed items in the generations set of claims. They then conduct experiments on synthetic and wikipedia data to confirm they claims.

**Strengths:**

- The paper proposes a level adaptive CP algorithm which to the best of my knowledge has not been done before in the LLM setting.
- The paper is clearly written (for someone who has published in CP before it was easy to follow)
- The paper tackles an important problem of hallucinations in LLMs which is timely and crucial for trustworthy LLMs
- The paper also proposes a boosted CP version and elegantly solves the new challenges albeit the main ideas were already introduced in Stutz. There is nonetheless novelty in the additional challenges to make it conditional. Prop 3.1 in particular is an interesting observation given the linearity
- The experiments overall show the superiority of the proposed methods compared to the current baselines. (there seems to be only one)

**Weaknesses:**

-  The paper starts off unfairly in figure 1, where the authors fixed a level of 90% for the baseline and 76% for the proposed method. This is misleading as one could easily also pick a lower coverage rate for the baseline and it would potentially look much less bad. This is concerning as it misleads the reader quite drastically until one reads the caption. Could you please explain why you chose 90% for the baseline but then only 80% for yours? Maybe i misunderstood your intentions with this figure.
- Secondly, I would like to note that the original conformal guarantees are finite sample however, if i understood correctly, this is no longer the case here eq (3), thm2.1. Hence my question is that are there any other baselines to compare against that are not conformal? I believe that this paper misses a substantial related works section.
- Thirdly, the experiments in the wikipedia dataset were choses with at most 3 wrong claims. Why did you choose 3  and not 0, 1 or 2 ? The fact that these experiments werent added in the appendix as ablation makes me question that 3 was chosen specifically to show the benefits of this paper. Please add these additional experiments for completeness as this number 3 was chosen out of nowhere. This also leads me to my next concern which is that of, even if we have at most 3 wrong claims, we can still not point to which one is wrong ... Could the authors please elaborate on the usefulness of this in the first place? Shouldnt you set the threshold to 0 wrong claims? This is a major concern from my side as this guarantee doesnt allow me to do anything useful in practice.
-  Fourthly, I also agree with the authors that the exchangeable assumption is a big drawback of this method in practice. The method that needs to be trained with the conditional boosting, is fixed for a given type data. This can be quite limiting.
- Lastly, there are not sufficient experimental results to support your claims. You have a single toy experiment with some Wiki dataset. Additional experiments or at least extensive ablation studies would be needed to convince me about the usefulness of this method.

**Questions:**

Please see above.
I am more than happy to raise my score if the above have been addressed.

**Limitations:**

Yes

---

> ### Author Rebuttal · Authors · 2024-08-06
>
> We thank for the reviewer for their constructive criticism, and we hope that this rebuttal can both address some of the limitations that they identified and certain misunderstandings.
>
> First, we would like to briefly note that the level-adaptive CP work is novel both in and outside of the LLM setting. In the more canonical regression setting, no previous work shows how to adapt the conformal guarantee at test-time to ensure moderate set sizes.
>
> 1) We respectfully disagree with the assertion that the paper starts off unfairly in Figure 1. The intent of that figure is to show that our method identifies the appropriate confidence level at test-time for preserving most of the output's claims. To further clarify, our method issues a guarantee that is different for each test output.  While it is true that running the conformal factuality method at 80\% would result in fewer claims being filtered for this example, this value would not work well universally over all test examples, e.g. other test examples may require 70\% or 90\% levels. Thus, the shortcoming of previous methods identified in this figure cannot be corrected by simply running those methods at a different level. One of the main contributions of our work is to develop a method that uses a data-driven approach to learn a desirable level for each individual test point and then issues a filtered response that is well-calibrated to the learned level. We apologize for any confusion caused by this figure. In our revision we will modify the text following Figure 1 in order to provide more context for why we believe this is an appropriate comparison.
>
> 2) We believe there may be some confusion here about the statements of our results. The theorems stated in our work are finite-sample exact guarantees that match the style and assumptions of those given in the conformal literature. For instance, note that while our result is more general, if $f$ is a constant function, the stated guarantee in Theorem 3.1 is identical to the typical coverage guarantee of split conformal prediction. We will aim to update the text surrounding our results to clarify this fact. We are unfamiliar with any non-conformal baselines for filtering claims from an LLM output with statistical guarantees (finite-sample valid or otherwise), but if any of the reviewers are aware of alternative approaches that we have not considered we would be happy to compare against them.
>
> 3) We agree that we ought to clarify our unconventional error guarantee. While we understand the reviewer's concerns, we chose a non-zero error guarantee because it is impossible to preserve most claims with this guarantee without reducing the level (value of $1-\alpha$) well below $50$\% (see Figure 3 in the rebuttal attachment). On the other hand, for $3$ errors, the level function we end up with outputs values that are similar to what we might typically associate with ``high probability guarantees'', i.e. $1-\alpha(X)$ is in the range of 80\%-95\%. We agree that the utility of a $3$ errors or fewer guarantee is less obvious than having $0$ errors, but if the probability of full correctness is undesirably low (maybe the user does not wish to issue a 40\% guarantee in order to preserve most claims), this trade-off may be worthwhile. Moreover, the intent of this experiment is to demonstrate the flexibility of our method to accommodate weaker error guarantees compared to exact factuality, not to suggest that $3$ errors is the desirable outcome for all applications. Overall, we agree with the reviewer that this choice requires further explanation in the main text, and we will revise the manuscript accordingly.
>
> 4) We agree that the exchangeability assumption is a limitation. We are not certain what the reviewer means by a given type of data, but in general we believe that the limitations of our method are no more significant than any existing approach to this problem. Additionally, as our methods target a form of conditional validity, they will provide robustness to some settings beyond exchangeability, e.g., when the test data undergoes a covariate shift.
>
> 5) Many of the reviewers inquired about additional experiments and we agree that the single real-data example given in the manuscript is in need of additional support. We have performed additional experiments that include the requested ablations as well as a large second experiment on a more real-world dataset involving medical question-answering. These results are shown in our global rebuttal to all reviewers and we hope that will assuage the reviewer's concerns.

---

> > ### Comment · Reviewer_tBPp · 2024-08-08
> > **Response**
> >
> > 1. Thanks for the clarification, however in practice one could simply pick alpha based on a heuristic i.e. i want to retain x% of the sentence and hence i can simply pick that. This would simple require a few easy heuristic steps from the calibration data which is on par with your additional need for compute to do boosting. What i am saying is this is purposefully taken to make baseline methods look bad. In practice if someone were to take 90% and see that they only retain so little claims, they would simply lower alpha. From this perspective can the authors please explain how this is not a trivial heuristic to compare against? Happy to keep this discussion in case i missed something, but the fact that you simply use 90% and then the adaptive method uses mostly around 70-80 is just misleading to me. I could easily find a fixed heuristic that does pretty much the same. I want to see concrete experiments where this slight adaptation towards the test sample benefits the end user in a significant manner.
> > 2. If it is truly finite sample, could the authors please tell me how the intervals scale with increased calibration data? The theorem has an expectation.
> > 3. The authors did not address why 3 was chosen except to make the results look good. I see in the attached pdf that you have results on the medical data that have 0 errors. I agree in settings where the model is presumably decent, this is achievable. In the Wikipedia dataset i assume that using 0 errors will make the whole method somewhat useless at below 50% guarantees. My main question is, if you say you have at most 3 errors, and cant tell the user which ones are the errors, what can the user possibly do with that information? The only use case is the 0 errors.
> > 4. Can the authors explain this covariate shift theory in more detail please. i might have missed that part.
> > 5. I thank the reviewer for the additional experiments. However as mentioned previsouly a simple heuristic of trying to pick the desired alpha based on how many sentences to retain would be just as usable. and hence my main question here is what the concrete advantages of this adaptive nature is. Looking at the histograms the nominal values seems to be unimodal. I am not saying to use this as a guide but a separate computation (heuristic compared to your adaptive method) where given some calibration data, we go over alpha [10, 20, ... 90%] and then pick the alpha which fits the desired retention level seems like a basic baselines to me.
> >
> > Happy to keep this discussion going as there might be some parts i might have missed or misunderstood. but i believe the above is still valid and would like the hear the authors answers to them before increasing the score.

---

> > > ### Author Response · Authors · 2024-08-09
> > > **Reply to 1-3**
> > >
> > > Thank you for the feedback and your openness to further discussion. We hope that these responses fully address your concerns.
> > >
> > > 1) One could certainly pick the fixed level of $1-\alpha$ based on a heuristic using the calibration data. For example, one may choose $1-\alpha$ to be the mean or median of the fitted function $1-\hat{\alpha}(X_{n+1})$. We will include these comparisons in the appendix. But as shown in the histograms attached to the global rebuttal, we find that $1-\hat{\alpha}(X_{n+1})$ takes on a large range of values (between 0.5-0.9 for the **MedLFQA** dataset and between 0.84-0.97 for the **FActscore** dataset). The only way to ensure good claim retention on all test examples would be to use the minimum value of $1-\hat{\alpha}(X_{n+1})$ (i.e. 0.5 for **MedLFQA** and 0.84 for **FActscore**). However, this would be needlessly liberal for many of the test points. Figure 6 on page 15 of the submission demonstrates that this level adaptivity is qualitatively important. The plotted points display the level at which 80\% of claims are retained; note that this level depends strongly on the number of views of the associated Wikipedia article. This figure shows that a single choice of level would not perform similarly to our method, and we will point the reader to this plot in order to emphasize this point in the main text.
> > >
> > >     One may also envision a simple theoretical setting in which test-time level adaptivity is required. Consider a two-groups model in which we have systematically higher claim scores for one group vs. another. We cannot issue guarantees at a constant level *and* preserve an equal number of claims in both groups.
> > >
> > >     Last, the goal of Figure 1 is to demonstrate that we cannot issue the typical "high probability" conformal guarantee while still retaining most claims. The final panel then shows what our method does. We prominently display the weaker guarantee of our method, so it is unclear to us what the reviewer means by misleading the reader.
> > >
> > > 2) The guarantee associated with conformal prediction holds in expectation over the calibration set (see: Proposition 1 in Shafer and Vovk's tutorial, Eq. 1 in Angelopoulos and Bates' Introduction). The previous work of Gibbs, Cherian, and Candes (2023), where the conditional targets we consider were originally introduced, shows that a larger calibration set allows the user to target a richer set of conditional guarantees, i.e. a richer function class $\mathcal{F}$. In particular, Figure 3 of that paper shows that if the calibration dataset is too small relative to the dimension of $\mathcal{F}$ the intervals (in a regression setting) or the number of filtered claims (in the LLM setting) will be quite large. Reviewer R2Vh also asked about this and in our revision we will look to update the article with a discussion that points readers to that reference.
> > >
> > >
> > > 3) The results look very similar (i.e., our methods are well-calibrated and give consistently above-target claim preservation) when the number of errors is chosen to be $0$. Unfortunately, OpenReview will not allow us to upload additional figures at this time, but we emphasize that we choose to display the $3$ error output primarily because we believe that a larger value for $1-\alpha$ would lead to a more interpretable statement for the reader. We disagree with the reviewer's claim that we chose this value because it makes our results look good. The guarantees of our method hold regardless; the low probability guarantee demonstrates that existing claim scoring functions are insufficiently accurate for this particular dataset, not that our method has significant flaws. In addition, there are many problems where we believe that such a guarantee is reasonable. Imagine we asked an LLM for a list of options, e.g., a list of $20$ restaurants suitable for a vegetarian. We are using the LLM to assemble a plausible list for further follow-up work, so some (though not too many) errors are tolerable.
> > >
> > >     We do not intend for this example to be a prescriptive description of how our method must be used, but rather an example of one possible application and loss function. The contribution of our paper is a framework for filtering claims from LLMs with rigorous guarantees.

---

> > > > ### Author Response · Authors · 2024-08-09
> > > > **Reply to 4-5**
> > > >
> > > > 4) In order to streamline the paper, we did not give an in-depth discussion of the connection between our guarantees and covariate shift. The work of Gibbs, Cherian, and Candes (2023), which introduces these conformal coverage targets extensively discusses the relationship between the conditional guarantees we consider and robustness to covariate shift. We will look to include information about this connection between our results and covariate shift robustness in our revision.\
> > > > For now, to understand this connection we remark that our method gives robustness to covariate shifts in which the training data $\\{(P_i,R_i,\\mathbf\{C\}\_i,\\mathbf\{W\}\_i)\\}\_{i=1}^n$ are sampled i.i.d. from a distribution $Q$, while the test point is sampled from the covariate shifted distribution $(P\_\{n+1\},R\_\{n+1\}) \sim f(P\_\{n+1\},R\_\{n+1\}) dQ\_\{(P,R)\}$ and $(\mathbf\{C\}\_\{n+1\},\mathbf\{W\}\_\{n+1\}) \sim Q_\{(\mathbf\{C\},\mathbf\{W\}) | (P,R)}$. Here, we use the notation $f(P\_\{n+1},R_{n+1\}) dQ\_\{(P,R\)}$ to refer to the distribution in which the covariate space is re-weighted by $f$. More precisely, using the subscript $f$ to denote the re-weighting, we have that for any set $A$,
> > > > $$
> > > > \\mathbb\{P\}\_f((P\_\{n+1\},R\_\{n+1\}) \in A) = \\frac{\\mathbb{E}\_Q[f((P,R)) \\mathbf\{1\}[(P,R) \in A]]\}{\\mathbb\{E\}\_Q[f(P,R)]\}.
> > > > $$
> > > > Using these definitions, our Theorem 3.1 has the immediate corollary that (under the assumptions and setting stated in the theorem),
> > > > $$
> > > > \\mathbb\{P\}\_f(L(\\hat\{F\}(\\mathbf\{C\}\_\{n+1\} ; \\hat\{\tau\}\_\{\\text\{rand\}\}(X\_\{n+1\}) ), \\mathbf\{W\}\_\{n+1\}) \\leq \\lambda ) = 1-\\alpha,
> > > > $$
> > > > for all non-negative functions $f \in \mathcal{F}$ with $0 <\mathbb{E}_Q[f(P,R)] < \infty$. This means that for all covariate shifts belonging to the class $\mathcal{F}$, the filtering procedure is guaranteed to control the loss with probability $1-\alpha$.
> > > >
> > > > 5) Following our response to point $1$, if we used e.g., the mode of the histogram, as a single fixed level for $\alpha$, the results would be unnecessarily liberal for many test points and would fail to retain a sufficient number of claims for others.

---

> > > > > ### Comment · Reviewer_tBPp · 2024-08-12
> > > > > **Response**
> > > > >
> > > > > 1. I thank the authors for their explanations. So to be clear, alpha is adaptive per sample and hence each sample will have a difference guarantee. In that case, could the authors please clarify what the exact technical contributions are compared to [Gibbs, Cherian, and Candes 2023]. Please elaborate the exact technical contribution as to me it looks like a simple extension, but i might be wrong here.
> > > > > 2. The authors said "The theorems stated in our work are finite-sample exact guarantees that match the style and assumptions of those given in the conformal literature." If something holds in expectation it is not finite sample ? Finite sample means that the theoretical guarantee is dependent on the number of calibration datapoints. I am just confused how you can call this finite sample and also "holds in expectation"? What i mean by finite sample is that there is a term such as 1/n in the upper/lower bound, as is present in the original conformal guarantee. Did i misunderstand something?
> > > > > 3. Here i believe we have to agree to disagree. I do not disagree that the guarantee still holds, what i am saying is that even if a guarantee holds the resulting output can be useless. This is like having a relatively bad classifier for Cifar10 and then outputting all 10 labels in the conformal set. The guarantee still holds but the output is useless, which is what i am suspecting will happen when you choose error = 0 in the experiments in the paper. However if that is not true, i would encourage the authors to add these experiments in the final version of the paper.
> > > > > 4. I do not believe i understand this, in such a condensed manner and hope the authors will add a thorough discussion in the final version of the paper.
> > > > > 5. Agree, i do see the appeal of adaptive scores now. please clarify for my 1. in terms of relationship with existing work.
> > > > >
> > > > > The rebuttal has convinced me to some extend, however i would like the authors to add these additional experiments, mentioned above, in the final version of their paper if the paper indeed get accepted. However, if the results turn out significantly different than the authors claimed in "The results look very similar (i.e., our methods are well-calibrated and give consistently above-target claim preservation) when the number of errors is chosen to be " that the authors will retract their paper.
> > > > >
> > > > > Hence i am willing to increase my score and after careful discussion with other reviewers would consider raising the score more.

---

> > > > > > ### Author Response · Authors · 2024-08-12
> > > > > > **Response**
> > > > > >
> > > > > > Thank you for your continued openness and engagement. We hope that these responses address your concerns.
> > > > > >
> > > > > > 1) The technical difference is that the previous method issues a fixed $1 - \alpha$ guarantee for each test point, while ours issues a data-dependent guarantee. We would first point out that this is itself a novel theoretical goal. It may be easiest to contextualize our contribution in the more canonical regression setting. Translating our results to this task, our work shows how to generate a calibrated prediction set whose size is consistently small. Independently of the LLM problem, our method presents a new approach to an active research area in conformal prediction: prediction set efficiency. Rather than designing conformity scores that we hope will lead to small prediction sets (which is not always possible), we can adjust the coverage guarantee to achieve this same goal.
> > > > > >
> > > > > >     Second, we do not believe that it is obvious how to define and achieve a meaningful level-adaptive guarantee. For example, another reviewer asked why we cannot simply run the conditional calibration procedure at the fixed level given by the $\alpha(\cdot)$ function estimated using the first data split. Unfortunately, this approach subtly breaks the exchangeability argument underpinning the proof of Gibbs, Cherian, and Candes. Our procedure circumvents this obstacle by estimating a data-dependent quantile for each calibration point, thereby preserving the symmetry of the algorithm. Achieving the calibration result we present in this paper requires us to not only modify the quantile regression presented in Gibbs, Cherian, and Candes, but also to think carefully about how one might design the function class used in their procedure. In our experiments, for example, we show that it is possible to add polynomials and group-indicators in $\alpha(\cdot)$ to $\mathcal{F}$ in order to achieve so-called smooth and binned calibration error guarantees.
> > > > > >
> > > > > > 2) We believe that the reviewer has misunderstood. Finite-sample does not require that the theoretical guarantee depends on the number of data points or that it must not hold in expectation. In particular, finite-sample guarantees can be exact (i.e., there is no error depending on $n$) and any guarantee that is stated in terms of a probability can be rewritten as an expectation over an indicator function.
> > > > > >
> > > > > >     The conformal guarantee that the reviewer refers to is $$1 - \alpha \leq \mathbb{P}(Y_{n + 1} \in \hat{C}(X_{n + 1})) \leq 1 - \alpha + \frac{1}{n + 1}.$$ Here the probability is taken over both the randomness in the test point and the calibration set used, i.e., it holds in expectation over both sets of random variables. However, a minor auxiliary randomization yields an exact finite-sample guarantee for conformal prediction, i.e., $\mathbb{P}(Y_{n + 1} \in \hat{C}\_\{\text\{rand.\}\}(X_{n + 1})) = 1 - \alpha$. Our procedure is also randomized to achieve an exact finite-sample guarantee (this is why you do not see any $1/n$ terms). It is possible to define a non-random construction for our quantile estimator, but we do not describe this method in the paper. Compared to the typical conformal guarantee, the upper bound would be modified from $1/(n + 1)$ to $\frac{d}{n + 1} \cdot \frac{1}{\mathbb{E}[\max_{i \in [n + 1]} f(X_{n + 1})]}$ where $d$ is the dimensionality of $\mathcal{F}$ (e.g., the number of groups). If $\mathcal{F}$ is chosen to be some collection of group-indicators, note that the second fraction in this upper bound is equal to $1$.
> > > > > >
> > > > > > 3) Though we agree that some users may not find the output of the $0$-error guarantee useful, we re-emphasize that the only degradation is in the nominal level offered (30-50\% instead of 80-95\%). Our method remains well-calibrated and retains ~70\% of the original claims. Even if the reviewer believes that relaxed error guarantees are not useful, we contend that there is value in honestly quantifying the high level of uncertainty inherent to issuing strong guarantees via deficient claim scoring methods.
> > > > > >
> > > > > > 4) We will do so. Apologies for any confusion.

---

### Official Review · Reviewer_R2Vh · 2024-07-12

**Soundness:** 3
**Presentation:** 3
**Contribution:** 4
**Rating:** 7
**Confidence:** 3

**Summary:**

The paper introduces two new conformal prediction methods aimed at enhancing the validity of large language models. The first method generalizes the conditional conformal procedure to issue weaker guarantees when necessary, thereby preserving utility. The second method improves the quality of the scoring function through a novel algorithm designed to differentiate the conditional conformal procedure. The proposed methods are validated using both synthetic and real-world datasets, addressing deficiencies in existing approaches by ensuring conditional validity and retaining valuable and accurate claims.

**Strengths:**

- The introduction of generalized conditional conformal procedures and improved scoring functions addresses critical gaps in current approaches.
- The paper is technically sound, with well-supported claims and robust experimental validation. The theoretical results, particularly Theorems 3.1 and 3.2, and Proposition 3.1, are well stated and proved. The methodology is well-developed, and the experimental design effectively demonstrates the efficacy of the proposed methods.
- The methods are evaluated on both synthetic and real-world datasets, effectively showcasing their practical utility and robustness.

**Weaknesses:**

- The methodology assumes the existence of an annotated calibration set of prompt-response-claim-annotation tuples, which may raise questions about the generalizability of the approach to different datasets or domains.
- Some readers may not be familiar with the metrics used to evaluate the method. Providing additional explanations for these metrics would enhance clarity and understanding.

**Questions:**

How sensitive is the methodology's performance to variations in the size and composition of the calibration set? How representative is this dataset of real-world large language model outputs?

**Limitations:**

The authors adequately address the limitations of their work in Section 5.

---

> ### Author Rebuttal · Authors · 2024-08-06
>
> We thank the reviewer for their positive comments and feedback. We address specific concerns below:
>
> 1) We agree that it is costly to obtain these tuples and it may not be feasible to do so in all cases. Nevertheless, given the large resources invested in LLM development by corporate labs, we do not believe that it is unrealistic that human-labels could be obtained for many real-world datasets of interest. Additionally, we hope that our work will help to provide a useful framework for further development in this field, and we have identified at least a few generation tasks where this methodology may prove helpful.
>
> 2) We are not certain which metrics the reviewer is referring to. In our revision, we will aim to ensure that the notions of calibration and retention utilized in our work are clearly defined.
>
> 3) The validity guarantee output by our method holds in expectation over the calibration set regardless of its size. However, it will be more variable conditional on the calibration set if the dataset is smaller or relevant subsets are underrepresented. In general, this method will perform best if the complexity of the factuality guarantee (i.e. the complexity of the function class $\mathcal{F}$) is not too large relative to the size of the calibration set. This leads to a model selection question that is analogues to what one faces in many standard regression problems. A detailed discussion of these issues can be found in Gibbs, Cherian, and Candes (2023). Since this article is primarily concerned with novel improvements to the methodology of Gibbs, Cherian, and Candes (2023) that enable its application to LLMs, we do not discuss these selection details in much depth here.
>
> 4) The Wikipedia dataset might appear synthetic since most people are not interested in writing biographies. Nevertheless, we believe it is still representative of the types of tasks LLMs may be used for. To help further demonstrate the range of possible applications of our method, we have performed additional experiments on a second medical question-answering dataset. We believe this example constitutes a highly relevant and realistic application for chat bots. Figures showing results from this experiment can be found in our global rebuttal to all reviewers. Overall, we find that our methods perform well on this example, obtaining both good calibration and improved claim retention.

---

> > ### Comment · Reviewer_R2Vh · 2024-08-09
> >
> > Thanks for the clarification. I remain positive about the work.

---

### Official Review · Reviewer_9DaN · 2024-07-13

**Soundness:** 3
**Presentation:** 3
**Contribution:** 3
**Rating:** 7
**Confidence:** 4

**Summary:**

The paper presents two new conformal inference methods for obtaining validity guarantees on large language model (LLM) outputs. The authors consider the task of filtering invalid claims from LLM responses to ensure high probability factuality guarantees on the filtered output. They discuss two limitations of existing approaches and how their methods improve upon them – (i) previous methods provide a marginal guarantee over a random test prompt instead of conditional validity, and (ii) existing methods may remove too many claims as the score function is imperfect. To improve upon these, the authors generalize the conditional conformal framework of [1] to adapt the filtering threshold that controls monotone risks and adjust the error rate adaptively (level-adaptive conformal prediction). The second method introduced in the paper termed conditional boosting enables automated discovery of superior claim scoring functions via differentiation through the conditional conformal algorithm i.e., finding conformity scores that allow retaining more claims in the LLM output while also ensuring validity. The authors perform experiments on synthetic and real-world datasets to demonstrate the performance of their methods and improvement upon existing approaches.
&nbsp;
&nbsp;

[1] I. Gibbs, J. J. Cherian, and E. J. Candès. Conformal prediction with conditional guarantees. arXiv preprint arXiv:2305.12616, 2023.

**Strengths:**

**Originality**: The main novel contribution is the conditional boosting procedure that allows automatic discovery of new scoring functions that enable higher claim retention via differentiation through the conditional conformal algorithm. While the paper leverages the conditional conformal procedure in Gibbs et al [1] and conformal factuality framework from Mohri and Hashimoto [2], the generalization of [1] to incorporate general losses and adaptive error rate is a novel and important contribution. To add, the previous work is adequately cited in the paper.


**Quality**: The submission is technically sound and the claims are well supported by theory. The experimental results empirically demonstrate the performance improvements of the methods introduced over previous approaches.


**Clarity**: The submission is well-written and clear for the most part. I appreciate the introduction to conditional conformal procedure of Gibbs et al. [1] in section 2.1. It would be helpful to organize the empirical analysis better to understand the setup and make performance improvements offered by different methods more clear.


**Significance**: The conformal inference methods proposed in the paper provide practically useful guarantees for LLM outputs. I believe the conditional validity formulation and evaluation are important contributions and the results will be of interest to the community.


**References**

[1]  I. Gibbs, J. J. Cherian, and E. J. Candès. Conformal prediction with conditional guarantees. arXiv preprint arXiv:2305.12616, 2023.
&nbsp;

[2] C. Mohri and T. Hashimoto. Language models with conformal factuality guarantees.  arXiv preprint arXiv:2402.10978, 2024.

**Weaknesses:**

The main weaknesses and scope for improvement are wrt the empirical evaluation:
1. Lack of ablations
- While the experiments include comparison with the split conformal calibration method of Mohri and Hashimoto [2] for the Wikipedia biographies dataset, I did not see comparison of the level-adaptive method with the fixed level conditional conformal procedure in [1] (where $\mathcal{F}$ can probably just be a linear combination of group indicators?). I believe this would also help understand the contribution of the level-adaptive method.
- How do different sub-claim scoring functions compare to each other in practice?

2.  Lack of justification behind the choice of score functions and function class $\mathcal{F}$
- Why is the conformity score chosen as naive absolute score value in Fig 4, 7? How would the results change if a better score function is selected? While the validity will always hold true, the empirical performance can give better practical insights (as is the intended goal of the paper).

3. Additional comment on organization of results: In its current form, it is hard to understand the experimental setup and variations of the method used for different figures. The experiments section can be organized better for greater clarity.


**References**

[1]  I. Gibbs, J. J. Cherian, and E. J. Candès. Conformal prediction with conditional guarantees. arXiv preprint arXiv:2305.12616, 2023.
&nbsp;

[2] C. Mohri and T. Hashimoto. Language models with conformal factuality guarantees.  arXiv preprint arXiv:2402.10978, 2024.

**Questions:**

1. It seems that both methods require an additional step of data splitting. Would this introduce high variance if there are not enough samples in some subgroups? (it would also be helpful to mention the number of samples in each group e.g. in Fig 2, where there are just 381 total test points)

2. Is there any specific reason why experiments were not performed on other real datasets studied in Mohri and Hashimoto?

3. Typo: shouldn't it be $Y_{n+1}$ in Theorem 2.1?

**Limitations:**

The authors discuss limitations in the Limitations section.

---

> ### Author Rebuttal · Authors · 2024-08-06
>
> We thank the reviewer for their feedback. We will look to prepare a revision that clarifies our experimental choices and provides additional experiments addressing the reviewer's concerns. We hope the following responses address the reviewer's concerns.
>
> Weaknesses:
> 1) We agree that this would be helpful, and we performed additional experiments with the requested ablations for both the Wikipedia and new medical question-answering experiment. Discussion of these results as well as associated figures can be found in our global rebuttal to all reviewers.
>
>     In our experiments, the frequency scoring method is most effective, and the improvements obtained by our method are robust to changes in the score. In our revision, we will prepare additional figures reproducing our results for alternative sub-claim scoring approaches.
>
> 2) The difference between the two methods in Figures 3, 4, and 7 may be less pronounced if a superior scoring function was selected. The goal of this experiment is not to show a comprehensive comparison, but instead to demonstrate the theoretical pitfalls of marginal boosting (i.e., conformal training). In general, if the scoring function is perfectly chosen, no boosting method (targeting marginal or conditional validity) substantially alters the performance of the method. This is also apparent in the Wikipedia example where the frequency scoring function far outperforms the other approaches and boosting simply recovers the performance of the best scoring approach. We believe that in many realistic scenarios, users do not have *a priori* access to a strong scoring function and thus, as this example demonstrates, conditional boosting can offer significant benefits over marginal boosting. This can be partially seen in our results on the Wikipedia and medical question-answering datasets where boosting produces a stronger ensemble of many weak scoring functions.
>
> 3) We apologize for the lack of clarity in the experimental set-up. We will aim to prepare a revision that more clearly organizes this information.
>
> Questions:
>
> 1) The results we prove hold in expectation over the calibration set. We agree that the calibration-conditional validity of the filtering method will be more variable if the calibration set is smaller (or similarly if there are fewer samples in some subgroups). We will add additional information about the sample sizes to the figures in our revision.
>
> 2) The primary reason we used the Wikipedia/FActscore dataset was because of the availability of high-quality synthetic annotations. Mohri and Hashimoto hand-label the correctness of each line of the response for $50$ examples. Unfortunately, running our method with any complex conditional guarantee on a such a small calibration set would lead to highly variable outcomes. Thus, we have chosen to generate synthetic ground-truth using GPT-4 with the original Wikipedia page passed in as additional context. Prior literature has validated that this method leads to an accurate factuality annotation for the FActscore benchmark. By contrast, we are unaware of any existing methods for obtaining synthetic ground truth on the other datasets considered by Mohri and Hashimoto. Nevertheless, we do agree that additional experimental validation would be helpful. As a result, we have also analyzed a medical long-form question answering dataset (released just prior to the submission deadline) where both expert and high-quality synthetic annotations have been made available.
>
> 3) We thank the reviewer for identifying this typo and will correct it.

---

> ### Comment · Reviewer_9DaN · 2024-08-11
>
> I thank the authors for addressing the questions and for sharing additional experiments. I look forward to the revision with suggested changes in future versions. I remain positive about the work and would like to keep my score as above.

---

### Official Review · Reviewer_CDbE · 2024-07-13

**Soundness:** 2
**Presentation:** 3
**Contribution:** 2
**Rating:** 6
**Confidence:** 4

**Summary:**

The paper provides a conformal prediction framework for the generation of hallucination risk-bound. The method can involve any loss function instead of the 0-1 loss as traditional conformal classification. They also provide a conditional (group) conformal prediction guarantee. They optimize the loss function with improved conformal prediction efficiency.

**Strengths:**

1. The paper is well-structured and well-written. The background and related work in Sec 2 can be helpful for readers not familiar with the background.
2. Conformal prediction for LMs can provide statistical guarantee on the generation risk, which is important for universal deployment of diverse LLMs.

**Weaknesses:**

1. [Generalization to alternative targets]: the rigorous proof is lacking about why the prediction set provided by lines 191 and 192 can lead to the conformal prediction guarantee. In line 181-187, the paper claims some assumptions on the loss function to be used. However, how these assumptions are used to derive the final valid conformal prediction guarantee is missing.

2. [Generalization to alternative targets]: conformal risk control methods [1] with a similar guarantee should be included in the discussion.

[1] Angelopoulos A N, Bates S, Fisch A, et al. Conformal risk control[J]. arXiv preprint arXiv:2208.02814, 2022.

3. [Generalization to alternative targets]: missing related work [2,3] on risk control of LLMs with conformal prediction
[2] Kang, Mintong, et al. "C-rag: Certified generation risks for retrieval-augmented language models." ICML 2024
[3] Quach, Victor, et al. "Conformal language modeling." ICLR 2024

4. [Level-adaptive conformal prediction]: Equation (8) use different $\alpha$ for the pinball loss (different $\alpha_i$), which can breaks the pinball loss formulation in Equation 4 and might also deteriorate the original conclusion of prediction coverage of $1-\alpha$. Do the authors have any justifications on it?

5. [Level-adaptive conformal prediction]: I think the designed level-adaptive conformal prediction is not that effective. If our goal is to use an adaptive level $\alpha$ for a particular example, why not directly do the conformal calibration with the particular $\alpha$ so that we can get the desired coverage?  I am confused on the final goal of the method in Sec 3.2. What the adaptivity here different from what I am thinking?

6. [Conditional boosting]: The conformal training method (reference [18] in the paper) optimizes $\theta$ in an end-to-end way, but the paper claims that it is unclear how to backpropagate through the estimated cutoff in Line 236. What makes the difference in the scenario?

**Questions:**

Please refer to the weakness part for concrete questions and concerns.

**Limitations:**

discussed in Sec 5

---

> ### Author Rebuttal · Authors · 2024-08-06
>
> We thank the reviewer for their positive comments and constructive feedback. In response to the reviewer's specific comments:
>
> 1) The proof of this result is given in Appendix A. Our assumption that $L(\emptyset, \cdot) = 0$ is necessary to ensure the score function, $S(\mathbf{C}_i, \mathbf{W}_i)$ given on line 191 is well-defined. On the other hand, our assumption that the loss is monotone in the filter is necessary to derive an equivalence between loss control and the dual variable whose distribution is analyzed in the proof of Theorem A.1. As stated in Appendix A.1, Theorem 3.1 then follows as a corollary of Theorem A.1. In particular, note that on Line 423 in the proof of Theorem A.1 we explicitly utilize our monotonicity assumption. In our revision we will add additional clarification to the main text of the manuscript discussing why these assumptions are necessary.
>
> 2) The "Conformal Risk Control" article of Angelopoulos et al. (2022) is referenced in our discussion of related work (see line 32). The approach taken to quantifying the uncertainty of LLM outputs taken in Angelopoulos et al. (2022) is quite different from our work and so we have not given an in-depth comparison. Nevertheless, we agree that the guarantees and high-level approach taken there are similar to our work and we will look to clarify this connection further in our revision.
>
> 3) We have referenced Quach et al. (2024) in our discussion of related work (see line 32). We thank the reviewer for bringing Kang et al. to our attention and will add this article to our discussion of related work in our revision. Like Quach et al., Kang et al. analyzes how many LLM generations are required to control a monotone risk. While the conformal methods used are related, we would emphasize that our approaches are both quantitatively and qualitatively different, i.e., we filter a single generation to obtain high-probability control of the loss.
>
> 4) Our goal is to adapt the coverage level of our method to the underlying difficulty of filtering falsehoods out of each example. To do this, we fit an adaptive function, $\alpha(X)$ in order to ensure that running our conformal method at level $1-\alpha(X_{n+1})$ will preserve at least a user-specified fraction of the LLM response. Then, we run an augmented quantile regression-type procedure in which data point $i$ is assigned loss $\ell_{\alpha(X_i)}(S(\mathbf{C}_i,\mathbf{W}_i) - g(X_i))$.
>
> 5) We are not entirely sure what exact alternative method the reviewer has in mind. Perhaps the reviewer is asking why we cannot simply use the value $\alpha^*(X_{n+1})$ such that when we run split conformal at this exact level, the desired percentage of claims are retained. Alternatively, perhaps the reviewer is interested in the method where we use $\alpha(X_{n+1})$ as a plug-in everywhere so that data point $i$ receives loss $\ell_{\alpha(X_{n+1})}(S(\mathbf{C}_i,\mathbf{W}_i) - g(X_i))$. Unfortunately, both these methods can incur selection bias and neither is valid in general. On a technical level, both of these approaches treat the test data point asymmetrically to the training data. This breaks the exchangeability of the data and thus does not prove a validity guarantee. On the other hand, our method, which indeed modifies the pinball loss, maintains exchangeability of the data and thus guarantees control of the loss. This result is rigorously proven in Theorem A.1.
>
> 6) The conformal training method in Stutz et al. (2021) backpropagates through the split conformal cutoff. Since the split conformal cutoff is defined to be a quantile of the calibration set, the value of cutoff is actually a particular calibration set score, i.e., $\text{cutoff} = S_{\lceil (n + 1) (1 - \alpha) \rceil}$. Differentiating with respect to some $\lambda$ that parameterizes the scoring function is then relatively straightforward as one may simply examine what happens to the score $S_{\lceil (n + 1) (1 - \alpha) \rceil}$ in isolation. It is worth noting that Stutz et al. (2021) use a "differentiable sorting" algorithm to obtain smoother gradients. By contrast, our cutoff is defined by the function of a solution to an optimization problem. The derivative of such an optimum is typically obtained via a second-order Taylor approximation (cf. influence functions). However, in our case the objective of the optimization problem is not differentiable and thus we cannot apply such an approximation. As we show in Proposition 3.1, this technical challenge can be overcome; despite the non-differentiability, we prove that the optimum remains locally linear in the parameter of interest.

---

> ### Comment · Reviewer_CDbE · 2024-08-11
> **Thanks for rebuttal**
>
> I would like to thank the authors for the clarifications.
>
> My concerns 1,3,4,5 are addressed, but for Q2 and Q6, I appreciate additional clarifications:
>
> Q2: In conformal risk control paper, the conformal guarantee is valid for any risk function and any parameterized generation algorithm. Therefore, the method can be exactly tailored for the problem here. Concretely, the algorithm can be exactly cutting off claims with low scores with the parameter as the threshold. By running conformal risk control, we can identify the sets of parameters (i.e., thresholds for cutting off) which achieve valid hallucination risk. In this sense, the contribution in sec 3.1 is not the first trial in the conformal literature and I would like that the authors clarify more on the novelties for Sec 3.1.
>
> Q6: I do not understand why the objective leads to additional challenges. The difference that paper made compared to standard conformal prediction is to use another adaptive loss function as the conformity scores. Why this will lead to a non-differentiability challenge? In particular, why is this not resolved with differentiable sorting as conformal training paper?

---

> > ### Author Response · Authors · 2024-08-12
> > **Further clarification**
> >
> > We thank the reviewer for their considered response and hope that the clarifications below address both of their questions.
> >
> > 1) We will revise Section 3.1 to precisely describe the similarities and differences between our work and previous methods. In short, conformal risk control generalizes split conformal prediction to **marginally** control a monotone loss. Analogously, Section 3.1 shows how to generalize the conditional conformal method (Gibbs, Cherian, and Candes 2023) in order to obtain **conditional** control of a monotone loss. We agree that one could apply conformal risk control to this problem, but the resulting guarantee would not match the one provided in Section 3.1.
> >
> > 2) We believe that the reviewer has misunderstood our contribution. Our method is not equivalent to running split conformal prediction on some adaptive choice of conformity score. Instead, given any pre-defined conformity score, our paper shows how to compute a conditionally valid and calibrated estimate of its quantile. For context, because it only targets marginal validity, split conformal prediction uses the sample quantile (i.e., a particular sorted conformity score) in place of our more sophisticated estimator.
> >
> >     Running a method analogous to conformal training requires us to differentiate our quantile estimator, but it is not obvious that this derivative exists. This is because our estimate is given by the solution to an optimization problem with a non-differentiable objective. Our contribution is to show that despite this challenge, our final quantile estimate is locally linear in the calibration scores.

---

> > > ### Comment · Reviewer_CDbE · 2024-08-13
> > > **Thanks for clarification**
> > >
> > > Thank the authors for clarification! After reading other reviews and thinking with the clarified points, I remain positive on the paper and raise score to 6.

---

### Author Rebuttal · Authors · 2024-08-06

We thank all the reviewers for their thoughtful comments and time spent considering our manuscript.

Two of the reviewers (CDbE and tBPp) inquired about additional experiments and, in particular, ablations demonstrating the contributions of each of our methods individually. Additionally, reviewer R2Vh asked if the Wikipedia biographies dataset considered in the paper is representative of real LLM use cases. To address these concerns, we have performed ablation studies on both the Wikipedia dataset presented in the manuscript and on a new dataset consisting of medical question-answering examples. In all cases, we find that our adaptive level and conditional boosting methods are effective in improving the percentage of claims retained both individually and in combination. Figures 1 and 2 in the attached document display the results of these new experiments.

The reviewers raise a number of additional questions about both the theoretical guarantees of our method and the parameter choices made in our experiments. We thank the reviewers for these comments. We will revise the manuscript to clarify each of these issues and to explain the reasoning by our experimental set-up in detail, e.g., Figure~3 in the attached document explains why we chose a relaxed error criterion for the FActscore experiment. In general, we find that the results of our experiments are robust to our choices. More detailed responses outlining our planned changes and addressing the specific concerns raised by the reviewers can be found in the individual rebuttals below.

---

### Decision · Program_Chairs · 2024-09-25

**Decision:**

Accept (poster)

**Comment:**

This paper presents conformal inference methods for obtaining validity guarantees on large language model (LLM) outputs, a problem known as conformal factuality (Mohri and Hashimoto 2024). It leverages previous work (Gibbs et al. 2023) to obtain **conditional** validity through level-adaptive conformal prediction. It also introduces a training method called "conditional boosting" which differentiates through the conditional conformal algorithm to obtain better conformity scores. Experiments on synthetic and real-world datasets demonstrate the performance of their methods and improvement upon existing approaches.

Most reviewers point out as strengths the originality and soundness of the proposed approach (particularly the conditional boosting procedure) and the non-trivial generalization of (Gibbs et al. 2023) to incorporate general losses and adaptive error rate. The main weaknesses are the limited discussion about alternative frameworks (such as conformal risk control), the limited empirical analysis (with lack of ablations and not very well organized experimental section), and somewhat insufficient discussion about the practical limitations of providing non-zero error guarantee in a conformal factuality framework with LLMs.

While I agree with the reviewer tBPp that a guarantee of three factual errors is in general not very useful in practice -- and I strongly recommend acknowledging this limitation more explicitly in the paper or presenting examples of situations where this can be useful to avoid overselling the practical usefulness of the method -- the other contributions of this paper are solid and potentially useful. The author rebuttal adds more experiments which will enhance the paper and alleviates some of the other concerns. I also urge the authors to include more details (as mentioned in the answer to CDbE) as what the advantages / disadvantages of their proposed framework are with respect to other proposed frameworks in the conformal prediction literature, such as conformal risk control.